

# Krotov: A Python implementation of Krotov's method for quantum optimal control

**Michael H. Goerz**[1*]**, Daniel Basilewitsch**[2]**, Fernando Gago-Encinas**[2]**,**
**Matthias G. Krauss**[2]**, Karl P. Horn**[2]**, Daniel M. Reich**[2]**, Christiane P. Koch**[2,3]

**1** U.S. Army Research Lab, Computational
and Information Science Directorate, Adelphi, MD 20783, USA
**2** Theoretische Physik, Universität Kassel, Heinrich-Plett-Str. 40, D-34132 Kassel, Germany
**3** Dahlem Center for Complex Quantum Systems and Fachbereich Physik,
Freie Universität Berlin, Arnimallee 14, 14195 Berlin, Germany

⋆ mail@michaelgoerz.net

## Abstract

We present a new open-source Python package, *krotov*, implementing the quantum optimal control method of that name. It allows to determine time-dependent external fields for a wide range of quantum control problems, including state-to-state transfer, quantum gate implementation and optimization towards an arbitrary perfect entangler. Krotov's method compares to other gradient-based optimization methods such as gradient-ascent and guarantees monotonic convergence for approximately time-continuous control fields. The user-friendly interface allows for combination with other Python packages, and thus high-level customization.



# 1   Introduction

Quantum information science has changed our perception of quantum physics from passive understanding to a source of technological advances [1]. By way of actively exploiting the two essential elements of quantum physics, coherence and entanglement, technologies such as quantum computing [2] or quantum sensing [3] hold the promise for solving computationally hard problems or reaching unprecedented sensitivity. These technologies rely on the ability to accurately perform quantum operations for increasingly complex quantum systems. Quantum optimal control allows to address this challenge by providing a set of tools to devise and implement shapes of external fields that accomplish a given task in the best way possible [4]. Originally developed in the context of molecular physics [5,6] and nuclear magnetic resonance [7,8], quantum optimal control theory has been adapted to the specific needs of quantum information science in recent years [4,9]. Calculation of optimized external field shapes for tasks such as state preparation or quantum gate implementation have thus become standard [4], even for large Hilbert space dimensions as encountered in e.g. Rydberg atoms [10,11]. Experimental implementation of the calculated field shapes, using arbitrary waveform generators, has been eased by the latter becoming available commercially. Successful demonstration of quantum operations in various experiments [4,12–20] attests to the level of maturity that quantum optimal control in quantum technologies has reached.

In order to calculate optimized external field shapes, two choices need to be made – about the optimization functional and about the optimization method. The functional consists of the desired figure of merit, such as a gate or state preparation error, as well as additional constraints, such as amplitude or bandwidth restrictions [4,9]. Optimal control methods in general can be classified into gradient-free and gradient-based algorithms that either evaluate the optimization functional alone or together with its gradient [4]. Gradient-based methods

typically converge faster, unless the number of optimization parameters can be kept small. Most gradient-based methods rely on the iterative solution of a set of coupled equations that include forward propagation of initial states, backward propagation of adjoint states, and the control update [4]. A popular representative of concurrent update methods is GRadient Ascent Pulse Engineering (GRAPE) [21]. Krotov's method, in contrast, requires sequential updates [5, 22]. This comes with the advantage of guaranteed monotonic convergence and obviates the need for a line search in the direction of the gradient [23]. While GRAPE is found in various software packages, there has not been an open source implementation of Krotov's method to date. Our package provides that missing implementation.

The choice of Python as an implementation language is due to Python's easy-to-learn syntax, expressiveness, and immense popularity in the scientific community. Moreover, the QuTiP library [24, 25] exists, providing a general purpose tool to numerically describe quantum systems and their dynamics. QuTiP already includes basic versions of other popular quantum control algorithms such as GRAPE and the gradient-free CRAB [26]. The Jupyter notebook framework [27] is available to provide an ideal platform for the interactive exploration of the `krotov` package's capabilities, and to facilitate reproducible research workflows.

The `krotov` package presented herein targets both students wishing to enter the field of quantum optimal control, and researchers in the field. By providing a comprehensive set of examples, we enable users of our package to explore the formulation of typical control problems, and to understand how Krotov's method can solve them. These examples are inspired by recent publications [28–33], and thus show the use of the method in the purview of current research. In particular, the package is not restricted to closed quantum systems, but can fully address open system dynamics, and thus aide in the development of Noisy Intermediate-Scale Quantum (NISQ) technology [34]. Optimal control is also increasingly important in the design of experiments [4, 12–20], and we hope that the availability of an easy-to-use implementation of Krotov's method will facilitate this further.

Large Hilbert space dimensions [10, 11, 35, 36] and open quantum systems [30] in particular require considerable numerical effort to optimize. Compared to the Fortran and C/C++ languages traditionally used for scientific computing, and more recently Julia [37], pure Python code usually performs slower by two to three orders of magnitude [38, 39]. Thus, for hard optimization problems that require several thousand iterations to converge, the Python implementation provided by the `krotov` package may not be sufficiently fast. In this case, it may be desirable to implement the entire optimization and time propagation in a single, more efficient (compiled) language. Our Python implementation of Krotov's method puts an emphasis on clarity, and the documentation provides detailed explanations of all necessary concepts, especially the correct time discretization, see Appendix A.3, and the possibility to parallelize the optimization. Thus, the `krotov` package can serve as a reference implementation, leveraging Python's reputation as "executable pseudocode", and as a foundation against which to test other implementations.

This paper is structured as follows: In Sec. 2, we give a brief overview of Krotov's method as it is implemented in the package. Based on a simple example, the optimization of a state-to-state transition in a two-level system, we describe the interface of the `krotov` package and its capabilities. Section 3 goes beyond that simple example to discuss how the `krotov` package can be used to solve some common, more advanced problems in quantum optimal control, involving complex-valued control fields, optimization of quantum gates in Hilbert or Liouville space, optimization over an ensemble of noise realizations, and use of non-convex functionals which occur e.g. in the optimization towards an arbitrary perfect entangler. Section 4 compares Krotov's method to other methods commonly used in quantum optimal control, in order to provide guidance on when use of the `krotov` package is most appropriate. Section 5 presents future perspectives, and Section 6 concludes. Appendix A defines and explains the

time-discretized update equation that underlies the implementation of Krotov's method. Appendix B gives a detailed technical specification of the optimization algorithm in pseudocode format, and analyzes the required numerical resources with respect to CPU time and memory. Appendices C and D contain installation instructions for the `krotov` package and link to its online documentation.

## 2 Overview of Krotov's method and the `krotov` package

### 2.1 The quantum control problem

Quantum optimal control methods formalize the problem of finding "control fields" that steer the time evolution of a quantum system in some desired way. For closed systems, described by a Hilbert space state $|\Psi(t)\rangle$, this time evolution is given by the Schrödinger equation,

$$\frac{\partial}{\partial t} |\Psi(t)\rangle = -\frac{i}{\hbar} \hat{H}(t) |\Psi(t)\rangle \,, \tag{1}$$

where the Hamiltonian $\hat{H}(t)$ depends on one or more control fields $\{\epsilon_l(t)\}$. We often assume the Hamiltonian to be linear in the controls,

$$\hat{H}(t) = \hat{H}_0 + \epsilon_1(t)\hat{H}_1 + \epsilon_2(t)\hat{H}_2 + \dots \,, \tag{2}$$

but non-linear couplings may also occur, for example when considering non-resonant multi-photon transitions. For open quantum systems described by a density matrix $\hat{\rho}(t)$, the Liouville-von-Neumann equation

$$\frac{\partial}{\partial t}\hat{\rho}(t) = \frac{1}{\hbar}\mathcal{L}(t)\hat{\rho}(t) \tag{3}$$

replaces the Schrödinger equation, with the (non-Hermitian) Liouvillian $\mathcal{L}(t)$. The most direct example of a control problem is a state-to-state transition. The objective is for a known quantum state $|\phi\rangle$ at time zero to evolve to a specific target state $|\phi^{\text{tgt}}\rangle$ at final time $T$, controlling, e.g. a chemical reaction [40]. Another example is the realization of quantum gates, the building blocks of a quantum computer. In this case, the states forming a computational basis must transform according to a unitary transformation [2], see Section 3.2. Thus, the control problem involves not just the time evolution of a single state, but a set of states $\{|\phi_k(t)\rangle\}$. Generalizing even further, each state $|\phi_k(t)\rangle$ in the control problem may evolve under a different Hamiltonian $\hat{H}_k(\{\epsilon_l(t)\})$, see Section 3.3.

Physically, the control fields $\{\epsilon_l(t)\}$ might be the amplitudes of a laser pulse for the control of molecular systems or trapped atom/ion quantum computers, radio-frequency fields for nuclear magnetic resonance, or microwave fields for superconducting circuits. When there are multiple independent controls $\{\epsilon_l(t)\}$ involved in the dynamics, these may correspond e.g., to different color lasers used in the excitation of a Rydberg atom, or different polarization components of an electric field.

The quantum control methods build on a rich field of classical control theory [41, 42]. This includes Krotov's method [43–46], which was originally formulated to optimize the soft landing of a spacecraft from orbit to the surface of a planet, before being applied to quantum mechanical problems [5, 22, 47–49]. Fundamentally, they rely on the variational principle, that is, the minimization of a functional $J[\{|\phi_k^{(i)}(t)\rangle\}, \{\epsilon_l^{(i)}(t)\}]$ that includes any required constraints via Lagrange multipliers. The condition for minimizing $J$ is then $\nabla_{\phi_k, \epsilon_l} J = 0$. In rare cases, the variational calculus can be solved in closed form, based on Pontryagin's maximum principle [42]. Numerical methods are required in any other case. These start from an initial guess control (or set of guess controls, if there are multiple controls), and calculate

an update to these controls that will decrease the value of the functional. The updated controls then become the guess for the next iteration of the algorithm, until the value of the functional is sufficiently small, or convergence is reached.

## 2.2 Optimization functional

Mathematically, Krotov's method, when applied to quantum systems [5,22], minimizes a functional of the most general form

$$J[\{|\phi_k^{(i)}(t)\rangle\}, \{\epsilon_l^{(i)}(t)\}] = J_T(\{|\phi_k^{(i)}(T)\rangle\}) + \sum_l \int_0^T g_a(\epsilon_l^{(i)}(t))\,\mathrm{d}t + \int_0^T g_b(\{\phi_k^{(i)}(t)\})\,\mathrm{d}t, \quad (4)$$

where the $\{|\phi_k^{(i)}(T)\rangle\}$ are the time-evolved initial states $\{|\phi_k\rangle\}$ under the controls $\{\epsilon_l^{(i)}(t)\}$ of the $i$'th iteration. In the simplest case of a single state-to-state transition, the index $k$ vanishes. For the example of a two-qubit quantum gate, $\{|\phi_k\rangle\}$ would be the logical basis states $|00\rangle$, $|01\rangle$, $|10\rangle$, and $|11\rangle$, all evolving under the same Hamiltonian $\hat{H}_k \equiv \hat{H}$. The sum over $l$ vanishes if there is only a single control. For open system dynamics, the states $\{|\phi_k\rangle\}$ may be density matrices.

The functional consists of three parts:

- A final time functional $J_T$. This is the "main" part of the functional, and we can usually think of $J$ as being an auxiliary functional in the optimization of $J_T$. The most straightforward final time functional for a simple state-to-state transition $|\phi\rangle \rightarrow |\phi^{\text{tgt}}\rangle$ is [50]

$$J_{T,\text{ss}} = 1 - \left|\left\langle \phi^{\text{tgt}} \mid \phi(T) \right\rangle\right|^2, \quad (5)$$

where $|\phi(T)\rangle$ is the time evolution of $|\phi\rangle$ to final time $T$. For a quantum gate $\hat{O}$, a typical functional is [50]

$$J_{T,\text{re}} = 1 - \frac{1}{N}\,\text{Re}\left[\sum_{k=1}^N \tau_k\right], \quad \text{with} \quad \tau_k = \left\langle \phi_k^{\text{tgt}} \mid \phi_k(T) \right\rangle, \quad |\phi_k^{\text{tgt}}\rangle = \hat{O}\,|\phi_k\rangle, \quad (6)$$

and $N$ being the dimension of the logical subspace, e.g. $N = 4$ and $\{|\phi_k\rangle\} = \{|00\rangle, |01\rangle, |10\rangle, |11\rangle\}$ for a two-qubit gate. The use of the real part in the functional implies that we care about the global phase of the achieved gate.

- A running cost on the control fields, $g_a$. The most commonly used expression (and the only one currently supported by the krotov package) is [50]

$$\begin{aligned} g_a(\epsilon_l^{(i)}(t)) &= \frac{\lambda_{a,l}}{S_l(t)}\left(\epsilon_l^{(i)}(t) - \epsilon_{l,\text{ref}}^{(i)}(t)\right)^2; \quad \epsilon_{l,\text{ref}}^{(i)}(t) = \epsilon_l^{(i-1)}(t) \\ &= \frac{\lambda_{a,l}}{S_l(t)}\left(\Delta\epsilon_l^{(i)}(t)\right)^2, \end{aligned} \quad (7)$$

with the inverse "step width" $\lambda_{a,l} > 0$, the "update shape" function $S_l(t) \in [0,1]$, and the control update

$$\Delta\epsilon_l^{(i)}(t) \equiv \epsilon_l^{(i)}(t) - \epsilon_l^{(i-1)}(t), \quad (8)$$

where $\epsilon_l^{(i-1)}(t)$ is the optimized control of the previous iteration – that is, the guess control of the current iteration ($i$).

- An optional state-dependent running cost, $g_b$. This may be used to encode time-dependent control targets [51,52], or to penalize population in a subspace [53]. The presence of a state-dependent constraint in the functional entails an inhomogeneous term in the backward propagation in the calculation of the control updates in each iteration of Krotov's method, see Appendix A, and is currently not supported by the `krotov` package. Penalizing population in a subspace can also be achieved through simpler methods that do not require a $g_b$, e.g., by using a non-Hermitian Hamiltonian to remove population from the forbidden subspace during the time evolution.

## 2.3 Iterative control update

Starting from the initial guess control $\epsilon_l^{(0)}(t)$, the optimized field $\epsilon_l^{(i)}(t)$ in iteration $i > 0$ is the result of applying a control update,

$$\epsilon_l^{(i)}(t) = \epsilon_l^{(i-1)}(t) + \Delta\epsilon_l^{(i)}(t).\tag{9}$$

Krotov's method is a clever construction of a particular $\Delta\epsilon_l^{(i)}(t)$ that ensures

$$J[\{|\phi_k^{(i)}(t)\rangle\}, \{\epsilon_l^{(i)}(t)\}] \leq J[\{|\phi_k^{(i-1)}(t)\rangle\}, \{\epsilon_l^{(i-1)}(t)\}].$$

Krotov's solution for $\Delta\epsilon_l^{(i)}(t)$ is given in Appendix A. As shown there, for the specific running cost of Eq. (7), using the guess control field $\epsilon_l^{(i-1)}(t)$ as the "reference" field, the update $\Delta\epsilon_l^{(i)}(t)$ is proportional to $\frac{S_l(t)}{\lambda_{a,l}}$. Note that this also makes $g_a$ proportional to $\frac{S_l(t)}{\lambda_{a,l}}$, so that Eq. (7) is still well-defined for $S_l(t) = 0$. The (inverse) Krotov step width $\lambda_{a,l}$ can be used to determine the overall magnitude of $\Delta\epsilon_l^{(i)}(t)$. Values that are too large will change $\epsilon_l^{(i)}(t)$ by only a small amount in every iteration, causing slow convergence. Values that are too small will result in numerical instability, see Appendix. A.3. The update shape function $S_l(t)$ allows to ensure boundary conditions on $\epsilon_l^{(i)}(t)$: If both the guess field $\epsilon_l^{(i-1)}(t)$ and $S_l(t)$ switch on and off smoothly around $t = 0$ and $t = T$, then this feature will be preserved by the optimization. A typical example for an update shape is

$$S_l(t) = \begin{cases} B(t; t_0 = 0, t_1 = 2t_{\text{on}}) & \text{for} \quad 0 < t < t_{\text{on}} \\ 1 & \text{for} \quad t_{\text{on}} \leq t \leq T - t_{\text{off}} \\ B(t; t_0 = T - 2t_{\text{off}}, t_1 = T) & \text{for} \quad T - t_{\text{off}} < t < T, \end{cases}\tag{10}$$

with the Blackman shape

$$B(t; t_0, t_1) = \frac{1}{2}\left(1 - a - \cos\left(2\pi\frac{t - t_0}{t_1 - t_0}\right) + a\cos\left(4\pi\frac{t - t_0}{t_1 - t_0}\right)\right), \quad a = 0.16,\tag{11}$$

which is similar to a Gaussian, but exactly zero at $t = t_0, t_1$. This is essential to maintain the typical boundary condition of zero amplitude at the beginning and end of the optimized control field. Generally, *any* part of the control field can be kept unchanged in the optimization by choosing $S_l(t) = 0$ for the corresponding intervals of the time grid.

## 2.4 Example: state-to-state transition in a two-level system

As a first taste of the `krotov` package's usage, we consider a simple but complete example for the optimization of a state-to-state optimization in Hilbert space, specifically the transformation $|0\rangle \rightarrow |1\rangle$ in a two-level system $\hat{H} = -\frac{\omega}{2}\hat{\sigma}_z + \epsilon(t)\hat{\sigma}_x$, where $\hat{\sigma}_z$ and $\hat{\sigma}_x$ are the Pauli-z and Pauli-x matrices, respectively, $\omega$ is the transition frequency between the levels $|0\rangle$ and $|1\rangle$, and

$\epsilon(t)$ is the control field. In the language of quantum computing, we are attempting to realize a bit-flip of a qubit from zero to one. The example assumes that the krotov package and other prerequisites have been installed on the user's system, see Appendix C. The full example script, as well as a Jupyter notebook version are also available as part of the package's online documentation, along with additional examples, see Appendix D.

```python
#!/usr/bin/env python
"""Example script for the optimization of a simple state-to-state
transition in a two-level system"""
import krotov
import qutip
import numpy as np

# First, we define the physical system (a simple TLS)

def hamiltonian(omega=1.0, ampl0=0.2):
    """Two-level-system Hamiltonian

    Args:
        omega (float): energy separation of the qubit levels
        ampl0 (float): constant amplitude of the driving field
    """
    H0 = -0.5 * omega * qutip.operators.sigmaz()
    H1 = qutip.operators.sigmax()

    def guess_control(t, args):
        return ampl0 * krotov.shapes.flattop(
            t, t_start=0, t_stop=5, t_rise=0.3, func="blackman"
        )

    return [H0, [H1, guess_control]]

H = hamiltonian()
tlist = np.linspace(0, 5, 500)

# Second, we define the control objective: a state-to-state
# transition from the |0⟩ eigenstate to the |1⟩ eigenstate

objectives = [
    krotov.Objective(
        initial_state=qutip.ket("0"), target=qutip.ket("1"), H=H
    )
]

# The magnitude of the pulse updates at each point in time are
# determined by the Krotov step size lambda_a and the
# time-dependent update shape (in [0, 1])
def S(t):
    """Shape function for the field update"""
    return krotov.shapes.flattop(
        t, t_start=0, t_stop=5, t_rise=0.3, func="blackman"
    )

# set required parameters for H[1][1] (the guess_control)
pulse_options = {H[1][1]: dict(lambda_a=5, update_shape=S)}
```

```python
56
57 # Before performing the optimization, it is usually a good idea
58 # to observe the system dynamics under the guess pulse. The
59 # mesolve method of the objective delegates to QuTiP's mesolve,
60 # and can calculate the expectation values of the projectors
61 # onto the |0⟩ and |1⟩ states, i.e., the population.
62
63 proj0, proj1 = (qutip.ket2dm(qutip.ket(l)) for l in ("0", "1"))
64 e_ops = [proj0, proj1]
65 guess_dynamics = objectives[0].mesolve(tlist, e_ops=e_ops)
66
67 # the resulting expectations values are in guess_dynamics.expect.
68 # The final-time populations are:
69
70 print(
71     "guess final time population in |0⟩, |1⟩: %.3f, %.3f\n"
72     % tuple([guess_dynamics.expect[l][-1] for l in (0, 1)])
73 )
74
75
76 # Now, we perform the actual optimization
77
78 opt_result = krotov.optimize_pulses(
79     objectives,
80     pulse_options=pulse_options,
81     tlist=tlist,
82     propagator=krotov.propagators.expm,
83     chi_constructor=krotov.functionals.chis_ss,
84     info_hook=krotov.info_hooks.print_table(
85         J_T=krotov.functionals.J_T_ss
86     ),
87     check_convergence=krotov.convergence.Or(
88         krotov.convergence.value_below('1e-3', name='J_T'),
89         krotov.convergence.check_monotonic_error,
90     ),
91     store_all_pulses=True,
92 )
93
94 print("\n", opt_result, sep='')
95
96
97 # We can observe the population dynamics under the optimized
98 # control
99
100 opt_dynamics = opt_result.optimized_objectives[0].mesolve(
101     tlist, e_ops=[proj0, proj1]
102 )
103
104 print(
105     "\noptimized final time population in |0⟩, |1⟩: %.3f, %.3f"
106     % (opt_dynamics.expect[0][-1], opt_dynamics.expect[1][-1])
107 )
```

The example starts by importing the `krotov` package, as well as QuTiP (the "Quantum Toolbox in Python") [24, 25] and NumPy (the standard package providing numeric arrays in Python) [54], used here to specify the propagation time grid. The integration of the `krotov` package with QuTiP is central: All operators and states are expressed as `qutip.Qobj` objects. Moreover, the `optimize_pulses` interface for Krotov's optimization method is inspired by the interface of QuTiP's central `mesolve` routine for simulating the system dynamics of a closed or open quantum system. In particular, when setting up an optimization, the (time-

dependent) system Hamiltonian should be represented by a nested list. That is, a Hamiltonian of the form $\hat{H} = \hat{H}_0 + \epsilon(t)\hat{H}_1$ is represented as H = [H0, [H1, eps]] where H0 and H1 are qutip.Qobj operators, and eps representing $\epsilon(t)$ is a function with signature eps(t, args), or an array of control values with the length of the time grid (tlist parameter). The hamiltonian function in line 11 of the example sets up exactly such an operator, using a control field with a flattop/Blackman envelope as specified in Eqs. (10, 11).

The next steps in the example set up the arguments required for the optimization initiated in line 78. The optimize_pulses function is the central routine provided by the krotov package. Its most important parameters are

- objectives: a list of objectives, each of which is an instance of krotov.Objective. Each objective has an initial_state, which is a qutip.Qobj representing a Hilbert space state or density matrix, a target (usually the target state that the initial state should evolve into when the objective is fulfilled), and a Hamiltonian or Liouvillian H in the nested-list format described above. In this example, there is a single objective for the transition $|0\rangle \rightarrow |1\rangle$ under the Hamiltonian initialized in line 29. The objectives express the goal of the optimization *physically*. However, they do not fully specify the functional $J_T$ that encodes the goal of the optimization *mathematically*: instead, $J_T$ is implicit in the chi_constructor argument, see below.

- pulse_options: a dictionary that maps each control to the parameters $\lambda_{a,l}$ (the Krotov update step size) and $S_l(t)$ (the update shape). In this example, H[1][1] refers to the guess_control in line 21. The value of 5 for $\lambda_a$ (no index $l$, as there is only a single control) was chosen by trial and error. $S(t)$ corresponds to the function defined in Eqs. (10, 11). The fact that $S(t)$ is the same formula as the envelope of the guess_control is incidental: $S(t)$ as the update_shape in the pulse_options only scales the *update* of the control field in each iteration, in this case enforcing that the value of the optimized fields remains zero at initial and final time.

- tlist: An array of time grid values in $[0, T]$. Internally, the controls are discretized as piecewise-constant on the intervals of this time grid. Here, the time grid is initialized in line 30, with 500 points between $t_0 = 0$ and $T = 5$. This is chosen such that the piecewise-constant approximation is sufficiently good to not affect the results within the shown precision of three significant digits.

- propagator: A routine that calculates the time evolution for a state over a single interval of the time grid. This allows the optimization to use arbitrary equations of motion. Also, since the main numerical effort in the optimization is the forward- and backward propagation of the states, the ability to supply a highly optimized propagator is key to numerical efficiency. In this example, we use the expm propagator that is included in the krotov package. It evaluates the result of the time propagation $|\Psi(t + dt)\rangle = \hat{U}|\Psi(t)\rangle$ by explicitly constructing the time evolution operator $\hat{U} = \exp[i\hat{H}\,dt]$ through matrix-exponentiation ($\hbar = 1$). Full matrix-exponentiation is inefficient for larger Hilbert space dimensions. For a dimension $> 10$ the expm propagator can still be useful as an "exact" propagator for debugging purposes.

- chi_constructor: a function that calculates a set of states $\{|\chi_k^{(i-1)}(T)\rangle\}$, according to the equation

$$\left|\chi_k^{(i-1)}(T)\right\rangle = -\left.\frac{\partial J_T}{\partial \langle\phi_k(T)|}\right|_{(i-1)}, \tag{12}$$

where the right-hand-side is evaluated for the set of states $\{|\phi_k^{(i-1)}(T)\rangle\}$ resulting from the forward-propagation of the initial states of the objectives under the guess con-

trols of iteration ($i$) – that is, the optimized controls of the previous iteration ($i-1$). The constructed states $\{|\chi_k^{(i-1)}(T)\rangle\}$ then serve as the boundary condition for the backward propagation in Krotov's method, see Appendices A, B. The `chi_constructor` implicitly defines the functional $J_T$: For every choice of functional, there is a corresponding `chi_constructor` that must be implemented from the analytic solution of Eq. (12). The `krotov` package includes the `chi_constructor` functions for the most common functionals in quantum optimal control. Here, `chis_ss` matches the functional $J_{T,\text{ss}}$ in Eq. (5),

$$
\begin{aligned}
|\chi^{(i-1)}(T)\rangle &= \frac{\partial}{\partial \langle\phi(T)|} \underbrace{\langle\phi(T)|\phi^{\text{tgt}}\rangle \langle\phi^{\text{tgt}}|\phi(T)\rangle}_{|\langle\phi^{\text{tgt}}|\phi(T)\rangle|^2}\Bigg|_{(i-1)} \\
&= \left(\langle\phi^{\text{tgt}}|\phi^{(i-1)}(T)\rangle\right)|\phi^{\text{tgt}}\rangle .
\end{aligned}
\tag{13}
$$

The call to `optimize_pulses` also includes two optional arguments that are used for convergence analysis. Without these, the optimization would run silently for a fixed number of iterations and then return a `Result` object (`opt_result` in the example) that contains the optimized controls discretized to the points of `tlist`, alongside other diagnostic data. The two parameters that allow to keep track of the optimization progress and to stop the optimization based on this progress, are

- `info_hook`: A function that receives the data resulting from an iteration of the algorithm, and may print arbitrary diagnostic information and calculate the value of the functional $J_T$. Any value returned from the `info_hook` will be available in the `info_vals` attribute of the final `Result` object. Here, we use an `info_hook` that prints a tabular overview of the functional values and the change in the functional in each iteration, see the script output below. This is the only place where $J_T$ is calculated *explicitly*, via the `J_T_ss` function that evaluates Eq. (5).

- `check_convergence`: A function that may stop the optimization based on the previously calculated `info_vals`. The `krotov` package includes suitable routines for detecting if the value of $J_T$, or the change $\Delta J_T$ between iterations falls below a specified limit. In the example, we chain two function via `Or`: The first function, `value_below`, stops the optimization when the value of $J_{T,\text{ss}}$ falls below $10^{-3}$, and the second function, `check_monotonic_error`, is a safety check to verify that the value of $J_{T,\text{ss}}$ decreases in each iteration. Both of these rely on the value of $J_{T,\text{ss}}$ having been calculated in the previous `info_hook`.

The parameter `store_all_pulses` is set here to ensure that the optimized controls from each iteration will be available in the `all_pulses` attribute of the `Result`, allowing for a detailed analysis of each iteration after the optimization ends, cf. Fig 1 below. Without this parameter, only the final optimized controls are kept. See the Jupyter notebook version of the example (Appendix D) for details on how to obtain Fig 1.

Before and after the optimization, the `mesolve` method of the `Objective` is used in the example to simulate the dynamics of the system under the guess control and the optimized control, respectively. This method delegates directly to QuTiP's `mesolve` function.

Overall, the example illustrates the general procedure for optimizing with the `krotov` package:

1. define the necessary quantum operators and states using QuTiP objects,

2. create a list of optimization objectives, as instances of `krotov.Objective`,

3. call `krotov.optimize_pulses` to perform an optimization of an arbitrary number of control fields over all the objectives.

Running the example script generates the following output:

```
guess final time population in |0⟩, |1⟩: 0.951, 0.049

iter.       J_T        ∫g_a(t)dt          J          ΔJ_T               Δ  secs
0      9.51e-01    0.00e+00    9.51e-01        n/a           n/a     1
1      9.24e-01    2.32e-03    9.27e-01    -2.70e-02    -2.47e-02     2
2      8.83e-01    3.53e-03    8.87e-01    -4.11e-02    -3.75e-02     2
3      8.23e-01    5.22e-03    8.28e-01    -6.06e-02    -5.54e-02     2
4      7.38e-01    7.39e-03    7.45e-01    -8.52e-02    -7.78e-02     1
5      6.26e-01    9.75e-03    6.36e-01    -1.11e-01    -1.01e-01     1
6      4.96e-01    1.16e-02    5.07e-01    -1.31e-01    -1.19e-01     1
7      3.62e-01    1.21e-02    3.74e-01    -1.34e-01    -1.22e-01     1
8      2.44e-01    1.09e-02    2.55e-01    -1.18e-01    -1.07e-01     2
9      1.53e-01    8.43e-03    1.62e-01    -9.03e-02    -8.19e-02     1
10     9.20e-02    5.80e-03    9.78e-02    -6.14e-02    -5.56e-02     1
11     5.35e-02    3.66e-03    5.72e-02    -3.85e-02    -3.48e-02     2
12     3.06e-02    2.19e-03    3.28e-02    -2.29e-02    -2.07e-02     1
13     1.73e-02    1.27e-03    1.86e-02    -1.33e-02    -1.20e-02     2
14     9.79e-03    7.24e-04    1.05e-02    -7.55e-03    -6.82e-03     2
15     5.52e-03    4.10e-04    5.93e-03    -4.27e-03    -3.86e-03     2
16     3.11e-03    2.31e-04    3.35e-03    -2.41e-03    -2.18e-03     2
17     1.76e-03    1.30e-04    1.89e-03    -1.36e-03    -1.23e-03     1
18     9.92e-04    7.36e-05    1.07e-03    -7.65e-04    -6.91e-04     1

Krotov Optimization Result
--------------------------
- Started at 2019-11-23 15:31:52
- Number of objectives: 1
- Number of iterations: 18
- Reason for termination: Reached convergence: J_T < 1e-3
- Ended at 2019-11-23 15:32:30 (0:00:38)

optimized final time population in |0⟩, |1⟩: 0.001, 0.999
```

The table that makes up the main part of the output is the result of the `print_table` function that was passed as an `info_hook` in line 84 of the example. The columns are the iteration number, where iteration 0 is an evaluation of the guess control; the value of the final time functional $J_T = J_{T,ss}$, see Eq. (5); the value of the running cost with $g_a(\epsilon_l^{(i)}(t))$ given by Eq. (7), which is a measure of how much the controls change in each iteration and thus allows to gauge convergence; the value of the total functional $J$ according to Eq. (4); the change in the value of $J_T$ relative to the previous iteration; the change in the total functional $J$; and finally the wallclock time in seconds spent on that iteration. The change in the total functional $\Delta J^{(i)}$ is guaranteed to be negative (monotonic convergence), up to the effects of time discretization. Note that

$$\Delta J^{(i)} = \Delta J_T^{(i)} + \sum_l \int_0^T \frac{\lambda_{a,l}}{S_l(t)} \left(\epsilon_l^{(i)}(t) - \epsilon_l^{(i-1)}(t)\right)^2 \, dt \neq J^{(i)} - J^{(i-1)} \tag{14}$$

for the values $J^{(i)}$, $J^{(i-1)}$ from two consecutive rows of the table. This is because $\Delta J^{(i)}$ must be evaluated with respect to a *single* reference field $\epsilon_{l,\text{ref}}^{(i)}(t)$ in Eq. (7), whereas the reported $J^{(i)}$ and $J^{(i-1)}$ use different reference fields, $\epsilon_l^{(i-1)}(t)$ and $\epsilon_l^{(i-2)}(t)$ respectively (the guess field in each iteration).

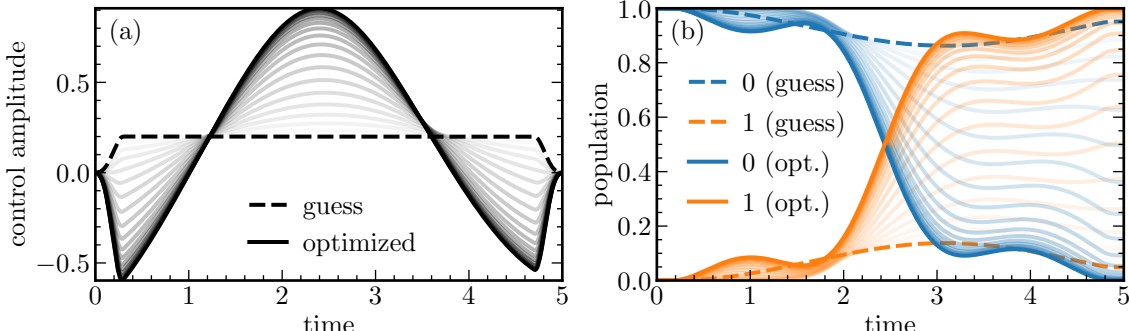

Figure 1: (Color online) Control fields and population dynamics for each iteration of the optimization procedure. (a) The initial guess control (dashed black line) and optimized controls (solid lines), with increasing opacity for each iteration of the optimization. The solid black line is the final optimized control. (b) The population dynamics for the two levels $|0\rangle$ (blue) and $|1\rangle$ (orange). The dynamics under the guess control are indicated by the dashed lines, and the dynamics under the optimized control of each iteration by the solid lines, with the opacity increasing with the iteration number. All quantities are in dimensionless units.

Figure 1 illustrates the progression of the optimization procedure. Panel (a) shows how the control field changes from the initial guess (dashed line) in each iteration. All these control fields are available in the `Result` object due to the parameter `store_all_pulses` in line 91 of the example. The optimized control fields are shown with increasing opacity for each iteration in the optimization. We can verify that the magnitude of the change in the control field in each iteration corresponds to the relative magnitude to the values in the column labeled $\int g_a(t)\,\mathrm{d}t$ in the output; as the optimization converges, the change in each iteration becomes smaller. All optimized controls preserve the boundary condition of a smooth switch-on/-off from zero at $t = t_0 = 0$ and $t = T = 5$, due to the choice of the `update_shape`. Panel (b) shows the population dynamics corresponding to the optimized control fields, obtained from plugging the optimized controls into the `objectives` and propagating the system with a call to the `mesolve` method. Again, the guess is indicated by the dashed line, and the opacity of the solid lines increases with the iteration number. We can verify the population transfer of only 0.049 under the guess control and the near perfect transfer ($\approx 0.999$) under the final optimized control.

## 3 Common optimization tasks

In the following, we discuss some of the most common tasks in quantum control and how they may be realized using the `krotov` package. The code snippets in this section are also available as complete examples in the form of interactive Jupyter notebooks in the Examples section of the online documentation, see Appendix D.

### 3.1 Complex-valued controls

When using the rotating wave approximation (RWA), it is important to remember that the target states are usually defined in the lab frame, not in the rotating frame. This is relevant for the construction of $|\chi_k(T)\rangle$. When doing a simple optimization, such as a state-to-state or a gate optimization, the easiest approach is to transform the target states to the rotating frame before calculating $|\chi_k(T)\rangle$. This is both straightforward and numerically efficient.

In the RWA, the control fields are usually complex-valued. In this case, the Krotov update equation is valid for both the real and the imaginary part independently. The most straightforward implementation of the method is to allow for real-valued controls only, requiring that any control Hamiltonian with a complex-valued control field is rewritten as two independent control Hamiltonians, one for the real part and one for the imaginary part of the control field. For example,

$$\epsilon^*(t)\hat{a} + \epsilon(t)\hat{a}^\dagger = \epsilon_{\mathrm{re}}(t)(\hat{a} + \hat{a}^\dagger) + \epsilon_{\mathrm{im}}(t)(i\hat{a}^\dagger - i\hat{a}), \tag{15}$$

with two independent control fields $\epsilon_{\mathrm{re}}(t) = \mathrm{Re}[\epsilon(t)]$ and $\epsilon_{\mathrm{im}}(t) = \mathrm{Im}[\epsilon(t)]$ with the control Hamiltonian $\hat{a} + \hat{a}^\dagger$ and $i\hat{a}^\dagger - i\hat{a}$, respectively.

### 3.2 Optimization towards a quantum gate

To optimize towards a quantum gate $\hat{O}$ in a *closed* quantum system, set one `Objective` for each state in the logical basis, with the basis state $|\phi_k\rangle$ as the `initial_state` and $|\phi_k^{\mathrm{tgt}}\rangle = \hat{O}|\phi_k\rangle$ as the `target`, cf. Eq. (6). The helper routine `gate_objectives` constructs the appropriate list of objectives, e.g. for a single-qubit Pauli-X gate:

```
objectives = krotov.gate_objectives(
    basis_states=[qutip.ket('0'), qutip.ket('1')],
    gate=qutip.operators.sigmax(),
    H=H,
)
```

The `gate_objectives` routine allows for open quantum systems as well. The parameter `liouville_states_set` indicates that the system dynamics are in Liouville space and defines the choice of an appropriate (minimal) set of matrices to track the optimization [30]. For example, to optimize for a $\sqrt{\mathrm{iSWAP}}$ gate in an open quantum system, three appropriately chosen density matrices $\hat{\rho}_1, \hat{\rho}_2, \hat{\rho}_3$ are sufficient to track the optimization progress [30]. Different emphasis can be put on each matrix, through relative weights 20:1:1 in the example below:

```
objectives = krotov.gate_objectives(
    basis_states=[qutip.ket(l) for l in ['00', '01', '10', '11']],
    gate=qutip.gates.sqrtiswap(),
    H=L,  # Liouvillian super-operator (qutip.Qobj instance)
    liouville_states_set='3states',
    weights=[20, 1, 1],
)
```

On many quantum computing platforms, applying arbitrary single-qubit gates is easy compared to entangling two-qubit gates. A specific entangling gate like CNOT is combined with single-qubit gates to form a universal set of gates. For a given physical system, it can be hard to know a-priori which entangling gates are easy or even possible to realize. For example, trapped neutral atoms only allow for the realization of diagonal two-qubit gates [30, 55] like CPHASE. However, the CPHASE gate is "locally equivalent" to CNOT: only additional single-qubit operations are required to obtain one from the other. A "local-invariants functional" [56] defines an optimization with respect to a such a local equivalence class, and thus is free to find the specific realization of a two-qubit gate that is easiest to realize. The objectives for such an optimization are generated by passing `local_invariants=True` to `gate_objectives`.

Generalizing the idea further, the relevant property of a gate is often its entangling power, and the requirement for a two-qubit gate in a universal set of gates is that it is a "perfect entangler". A perfect entangler can produce a maximally entangled state from a separable input state. Since 85% of all two-qubit gates are perfect entanglers [57, 58], a functional that targets an arbitrary perfect entangler [31, 32] solves the control problem with the least

constraints. The objectives for this optimization are initialized by passing `gate='PE'` to `gate_objectives`. Both the optimization towards a local equivalence class and an arbitrary perfect entangler may require use of the second-order update equation, see Sec. 3.4.

### 3.3 Ensemble optimization as a way to ensure robust controls

Control fields can be made robust with respect to variations in the system by performing an "ensemble optimization" [29]. The idea is to sample a representative selection of possible system Hamiltonians, and to optimize over an average of the entire ensemble. In the functional, Eq. (4), respectively the update Eq. (28), the index $k$ now numbers not only the states, but also different ensemble Hamiltonians: $\hat{H}(\{\epsilon_l(t)\}) \to \{\hat{H}_k(\{\epsilon_l(t)\})\}$.

The example considered in Ref. [29] is that of a CPHASE two-qubit gate on trapped Rydberg atoms. Two classical fluctuations contribute significantly to the gate error: deviations in the pulse amplitude ($\Omega = 1$ ideally), and fluctuations in the energy of the Rydberg level ($\Delta_{\mathrm{ryd}} = 0$ ideally). We also take into account decay and dephasing, and thus optimize in Liouville space, setting the objectives as in Sec. 3.2:

```
from math import pi  # standard library
objectives = krotov.gate_objectives(
    basis_states=[
        qutip.ket(l) for l in ['00', '01', '10', '11']
    ],
    gate=qutip.gates.cphase(pi),
    H=L(omega=1, delta=0),
    liouville_states_set='3states',
    weights=[0, 1, 1]
)
```

This will result in a list of two objectives for the density matrices $\hat{\rho}_2$ and $\hat{\rho}_3$ defined in Ref. [30]. The state $\hat{\rho}_1$ is omitted by setting its weight to zero, as the target gate is diagonal. The function L is assumed to return the Liouvillian for the system with given values for $\Omega$ and $\Delta_{\mathrm{ryd}}$.

An appropriate set of ensemble objectives (extending the objectives defined above) can now be generated with the help of the `ensemble_objectives` function.

```
import itertools  # standard library
ensemble_liouvillians = [
    L(omega, delta)
    for (omega, delta)
    in itertools.product(omega_vals, delta_vals)
]
objectives = krotov.objectives.ensemble_objectives(
    objectives, ensemble_liouvillians
)
```

Here `omega_vals` and `delta_vals` is assumed to contain values sampling the space of perturbed values $\Omega \neq 1$ and $\Delta_{\mathrm{ryd}} \neq 0$. For $M-1$ `ensemble_liouvillians`, i.e. $M$ systems including the original unperturbed system, the above call results in a list of $2M$ `objectives`. Note that all elements of `ensemble_liouvillians` share the same control pulses. As shown in Ref. [30], an optimization over the average of all these objectives via the functional in Eq. (6) results in controls that are robust over a wide range of system perturbations.

### 3.4 Optimization of non-linear problems or non-convex functionals

In Refs. [31, 32], a non-convex final-time functional for the optimization towards an arbitrary perfect entangler is considered. In order to guarantee monotonic convergence, the Krotov update equation must be constructed to second order, see Appendix A.2. In practice, this

means we must specify a scalar function $\sigma(t)$ that serves as the coefficient to the second order contribution.

For this specific example, a suitable choice is

$$\sigma(t) \equiv -\max(\varepsilon_A, 2A + \varepsilon_A), \tag{16}$$

where $\varepsilon_A$ is a small non-negative number. The optimal value for $A$ in each iteration can be approximated numerically as [22]

$$A = \frac{\sum_{k=1}^{N} 2\,\mathrm{Re}\left[\langle \chi_k(T)|\Delta\phi_k(T)\rangle\right] + \Delta J_T}{\sum_{k=1}^{N} |\Delta\phi_k(T)|^2}, \tag{17}$$

with

$$\Delta J_T \equiv J_T(\{\phi_k^{(i)}(T)\}) - J_T(\{\phi_k^{(i-1)}(T)\}). \tag{18}$$

In the `krotov` package, in order to make use of the second order contribution to the pulse update, we pass a parameter `sigma` to the `optimize_pulses` function:

```python
class sigma(krotov.second_order.Sigma):
    def __init__(self, A, epsA=0):
        self.A = A
        self.epsA = epsA

    def __call__(self, t):
        return -max(self.epsA, 2 * self.A + self.epsA)

    def refresh(
        self, forward_states, forward_states0,
        chi_states, chi_norms, optimized_pulses,
        guess_pulses, objectives, result,
    ):
        try:
            # info_vals contains values of PE functional
            Delta_J_T = (
                result.info_vals[-1][0] - result.info_vals[-2][0]
            )
        except IndexError:  # first iteration
            Delta_J_T = 0
        self.A = krotov.second_order.numerical_estimate_A(
            forward_states, forward_states0, chi_states,
            chi_norms, Delta_J_T
        )

oct_result = krotov.optimize_pulses(
    objectives,
    pulse_options=pulse_options,
    tlist=tlist,
    propagator=krotov.propagators.expm,
    chi_constructor=chi_constructor,  # from weylchamber package
    info_hook=calculate_PE_val,
    sigma=sigma(A=0.0),
)
```

The function `krotov.second_order.numerical_estimate_A` implements Eq. (17). The function defined by the instantiated `sigma` is used for the pulse update, and then the internal parameter, $A$ in this case, is automatically updated at the end of each iteration, via the `sigma`'s `refresh` method.

Even when the second order update equation is mathematically required to *guarantee* monotonic convergence, often an optimization with the first-order update equation (28) will

give converging results. Since the second order update requires more numerical resources (calculation and storage of the states $|\Delta\phi_k(t)\rangle$, see Appendix B), it is advisable to attempt an optimization with the first-order update equation first, and to only use the second order when the first order proves insufficient.

# 4 Comparison of Krotov's method and other optimization methods

In the following, we compare Krotov's method to other numerical optimization methods that have been used widely in quantum control, with an emphasis on methods that have been implemented as open source software. We first discuss iterative schemes derived from general variational calculus in Section 4.1 before making the connection to Krotov's method in particular, in Section 4.2. We then compare with GRadient Ascent Pulse Engineering (GRAPE) [4, 21] in Section 4.3, before highlighting the differences with gradient-free methods in Section 4.4. Finally, Section 4.5 provides some guidance for the choice of an appropriate optimization method for particular circumstances.

## 4.1 Iterative schemes from variational calculus

Gradient-based optimal control methods derive the condition for the optimal control field from the application of the variational principle to the optimization functional in Eq. (4). Since the functional depends both on the states and the control field, it is necessary to include the equation of motion (Schrödinger or Liouville-von-Neumann) as a constraint. That is, the states $\{|\phi_k\rangle\}$ must be compatible with the equation of motion under the control fields $\{\epsilon_l(t)\}$. In order to convert the constrained optimization problem into an unconstrained one, the equation of motion is included in the functional with the co-states $|\chi_k(t)\rangle$ as Lagrange multipliers [59–62].

The necessary condition for an extremum becomes $\delta J = 0$ for this extended functional. Evaluation of the extremum condition results in [62]

$$\Delta\epsilon_l(t) \propto \frac{\delta J}{\delta\epsilon_l} \propto \mathrm{Im}\langle\chi_k(t)|\hat{\mu}|\phi_k(t)\rangle, \tag{19}$$

where $\hat{\mu} = \partial\hat{H}/\partial\epsilon_l(t)$ is the operator coupling to the field $\epsilon_l(t)$. Equation (19) is both continuous in time and implicit in $\epsilon_l(t)$ since the states $|\phi_k(t)\rangle$, $|\chi_k(t)\rangle$ also depend on $\epsilon_l(t)$. Numerical solution of Eq. (19) thus requires an iterative scheme and a choice of time discretization.

The most intuitive time-discretization yields a *concurrent* update scheme [5, 47, 62],

$$\Delta\epsilon_l^{(i)}(t) \propto \mathrm{Im}\langle\chi_k^{(i-1)}(t)|\hat{\mu}|\phi_k^{(i-1)}(t)\rangle. \tag{20}$$

Here, at iterative step $(i)$, the backward-propagated co-states $\{|\chi_k(t)\rangle\}$ and the forward-propagated states $\{|\phi_k(t)\rangle\}$ both evolve under the 'guess' controls $\epsilon_l^{(i-1)}(t)$ of that iteration. Thus, the update is determined entirely by information from the previous iteration and can be evaluated at each point $t$ independently. However, this scheme does not guarantee monotonic convergence, and requires a line search to determine the appropriate magnitude of the pulse update [62].

A further ad-hoc modification of the functional [63] allows to formulate a family of update schemes that do guarantee monotonic convergence [64, 65]. These schemes introduce separate fields $\{\epsilon_l(t)\}$ and $\{\tilde{\epsilon}_l(t)\}$ for the forward and backward propagation, respectively, and

use the update scheme [66]

$$\epsilon_l^{(i)}(t) = (1-\delta)\tilde{\epsilon}_l^{(i-1)}(t) - \frac{\delta}{\alpha} \operatorname{Im}\left\langle \chi_k^{(i-1)}(t) \middle| \hat{\mu} \middle| \phi_k^{(i)}(t) \right\rangle, \tag{21a}$$

$$\tilde{\epsilon}_l^{(i)}(t) = (1-\eta)\epsilon_l^{(i-1)}(t) - \frac{\eta}{\alpha} \operatorname{Im}\left\langle \chi_k^{(i)}(t) \middle| \hat{\mu} \middle| \phi_k^{(i)}(t) \right\rangle, \tag{21b}$$

with $\delta, \eta \in [0,2]$ and an arbitrary step width $\alpha$. For the control of wavepacket dynamics, an implementation of this generalized class of algorithms is available in the WavePacket Matlab package [67].

## 4.2 Krotov's method

The method developed by Krotov [43–46] and later translated to the language of quantum control by Tannor and coworkers [5, 22, 47–49] takes a somewhat unintuitive approach to disentangle the interdependence of field and states by adding a zero to the functional. This allows to *construct* an updated control field that is guaranteed to lower the value of the functional, resulting in monotonic convergence. The full method is described in Appendix A, but its essence can be boiled down to the update in each iteration $(i)$, Eq. (8), taking the form

$$\Delta\epsilon_l^{(i)}(t) \propto \operatorname{Im}\left\langle \chi_k^{(i-1)}(t) \middle| \hat{\mu} \middle| \phi_k^{(i)}(t) \right\rangle, \tag{22}$$

with co-states $|\chi_k(t)^{(i-1)}\rangle$ backward-propagated under the *guess* controls $\{\epsilon_l^{(i-1)}(t)\}$ and the states $|\phi_k^{(i)}(t)\rangle$ forward-propagated under the *optimized* controls $\{\epsilon_l^{(i)}(t)\}$. Compared to the *concurrent* form of Eq. (20), the Krotov update scheme is *sequential*: The update at time $t$ depends on the states forward-propagated using the updated controls at all previous times, see Appendix A.3 for details.

It is worth noting that the sequential update can be recovered as a limiting case of the monotonically convergent class of algorithms in Eq. (21), for $\delta = 1$, $\eta = 0$. This may explain why parts of the quantum control community consider *any* sequential update scheme as "Krotov's method" [68, 69]. However, following Krotov's construction [43–46] requires no ad-hoc modification of the functional and can thus be applied more generally. In particular, as discussed in Section 3.4 and Appendix A.2, a second-order construction can address non-convex functionals.

In all its variants [5, 22, 47–49], Krotov's method is a first-order gradient with respect to the control fields (even in the second-order construction which is second order only with respect to the states). As the optimization approaches the optimum, this gradient can become very small, resulting in slow convergence. It is possible to extend Krotov's method to take into account information from the quasi-Hessian [23]. However, this "K-BFGS" variant of Krotov's method is a substantial extension to the procedure as described in Appendix B, and is currently not supported by the krotov package.

The update Eq. (22) is specific to the running cost in Eq. (7). In most of the schemes derived from variational calculus, cf. Section 4.1, a constraint on the *pulse fluence* is used instead. Formally, this is also compatible with Krotov's method, by choosing $\epsilon_{l,\mathrm{ref}}^{(i)}(t) \equiv 0$ in Eq. (7) [70]. It turns the *update* equations (22, 20) into *replacement* equations, with $\epsilon_l^{(i)}(t)$ on the left-hand side instead of $\Delta\epsilon_l^{(i)}(t)$, cf. Eq. (21) for $\delta = 1$, $\eta = 0$. In our experience, this leads to numerical instability and should be avoided. A mixture of *update* and *replacement* is possible when a penalty of the pulse fluence is necessary [71].

## 4.3 GRadient Ascent Pulse Engineering (GRAPE)

While the monotonically convergent methods based on variational calculus must "guess" the appropriate time discretization, and Krotov's method finds the sequential time discretization by

a clever construction, the GRAPE method sidesteps the problem by discretizing the functional *first*, before applying the variational calculus.

Specifically, we consider the piecewise-constant discretization of the dynamics onto a time grid, where the final time states $\{|\phi_k^{(i-1)}(T)\rangle\}$ resulting from the time evolution of the initial states $\{|\phi_k\rangle\}$ under the guess controls $\epsilon_n^{(i-1)}$ in iteration $(i)$ of the optimization are obtained as

$$|\phi_k^{(i-1)}(T)\rangle = \hat{U}_{N_T}^{(i-1)}\ldots\hat{U}_n^{(i-1)}\ldots\hat{U}_1^{(i-1)}|\phi_k\rangle, \tag{23}$$

where $\hat{U}_n^{(i-1)}$ is the time evolution operator on the time interval $n$ in Hilbert space,

$$\hat{U}_n^{(i-1)} = \exp\left[-\frac{\mathrm{i}}{\hbar}\hat{H}\big(\underbrace{\epsilon^{(i-1)}(\tilde{t}_{n-1})}_{\epsilon_n^{(i-1)}}\big)\mathrm{d}t\right]; \qquad \tilde{t}_n \equiv t_n + \mathrm{d}t/2. \tag{24}$$

The independent control parameters are now the scalar values $\epsilon_n$, respectively $\epsilon_{ln}$ if there are multiple control fields indexed by $l$.

The GRAPE method looks at the direct gradient $\partial J/\partial\epsilon_n$ and updates each control parameter in the direction of that gradient [21]. The step width must be determined by a line search.

Typically, only the final time functional $J_T$ has a nontrivial gradient. For simplicity, we assume that $J_T$ can be expressed in terms of the complex overlaps $\{\tau_k\}$ between the target states $\{|\phi_k^{\mathrm{tgt}}\rangle\}$ and the propagated states $\{|\phi_k(T)\rangle\}$, as e.g. in Eqs. (5, 6). Using Eq. (23) leads to

$$\begin{aligned}
\frac{\partial\tau_k}{\partial\epsilon_n} &= \frac{\partial}{\partial\epsilon_n}\langle\phi_k^{\mathrm{tgt}}|\hat{U}_{N_T}^{(i-1)}\ldots\hat{U}_n^{(i-1)}\ldots\hat{U}_1^{(i-1)}|\phi_k\rangle \\
&= \underbrace{\langle\phi_k^{\mathrm{tgt}}|\hat{U}_{N_T}^{(i-1)}\ldots\hat{U}_{n+1}^{(i-1)}}_{\langle\chi_k^{(i-1)}(t_{n+1})|}\frac{\partial\hat{U}_n^{(i-1)}}{\partial\epsilon_n}\underbrace{\hat{U}_{n-1}^{(i-1)}\ldots\hat{U}_1^{(i-1)}|\phi_k\rangle}_{|\phi_k^{(i-1)}(t_n)\rangle}
\end{aligned} \tag{25}$$

as the gradient of these overlaps. The gradient for $J_T$, respectively $J$ if there are additional running costs then follows from the chain rule. The numerical evaluation of Eq. (25) involves the backward-propagated states $|\chi_k(t_{n+1})\rangle$ and the forward-propagated states $|\phi_k(t_n)\rangle$. As only states from iteration $(i-1)$ enter in the gradient, GRAPE is a *concurrent* scheme.

The comparison of the sequential update equation (22) of Krotov's method and the concurrent update equation (20) has inspired a sequential evaluation of the "gradient", modifying the right-hand side of Eq. (25) to $\langle\chi_k^{(i-1)}(t_{n+1})|\partial_\epsilon U_n^{(i-1)}|\phi_k^{(i)}(t_n)\rangle$. That is, the states $\{|\phi_k(t)\rangle\}$ are forward-propagated under the optimized field [72]. This can be generalized to "hybrid" schemes that interleave concurrent and sequential calculation of the gradient [69]. An implementation of the concurrent/sequential/hybrid gradient is available in the DYNAMO Matlab package [69]. The sequential gradient scheme is sometimes referred to as "Krotov-type" [69, 73]. To avoid confusion with the specific method defined in Appendix A, we prefer the name "sequential GRAPE".

GRAPE does not give a guarantee of monotonic convergence. As the optimization approaches the minimum of the functional, the first order gradient is generally insufficient to drive the optimization further [23]. To remedy this, a numerical estimate of the Hessian $\partial^2 J_T/\partial\epsilon_j\partial\epsilon_{j'}$ should also be included in the calculation of the update. The L-BFGS-B quasi-Newton method [74, 75] is most commonly used for this purpose, resulting in the "Second-order GRAPE" [76] or "GRAPE-LBFGS" method. L-BFGS-B is implemented as a Fortran library [75] and widely available, e.g. wrapped in optimization toolboxes like SciPy [77]. This means that it can be easily added as a "black box" to an existing gradient optimization. As

a result, augmenting GRAPE with a quasi-Hessian is essentially "for free". Thus, we always mean GRAPE to refer to GRAPE-LBFGS. Empirically, GRAPE-LBFGS *usually* converges monotonically.

Thus, for (discretized) time-continuous controls, both GRAPE and Krotov's method can generally be used interchangeably. Historically, Krotov's method has been used primarily in the control of molecular dynamics, while GRAPE has been popular in the NMR community. Some potential benefits of Krotov's method compared to GRAPE are [23]:

- Krotov's method mathematically guarantees monotonic convergence in the continuous-time limit. There is no line search required for the step width $1/\lambda_{a,l}$.

- The sequential nature of Krotov's update scheme, with information from earlier times entering the update at later times within the same iteration, results in faster convergence than the concurrent update in GRAPE [69,78]. This advantage disappears as the optimization approaches the optimum [23].

- The choice of functional $J_T$ in Krotov's method only enters in the boundary condition for the backward-propagated states, Eq. (12), while the update equation stays the same otherwise. In contrast, for functionals $J_T$ that do not depend trivially on the overlaps [79–83], the evaluation of the gradient in GRAPE may deviate significantly from its usual form, requiring a problem-specific implementation from scratch. This may be mitigated by the use of automatic differentiation in future implementations [84,85].

GRAPE has a significant advantage if the controls are not time-continuous, but are *physically* piecewise constant ("bang-bang control"). The calculation of the GRAPE-gradient is unaffected by this, whereas Krotov's method can break down when the controls are not approximately continuous. QuTiP contains an implementation of GRAPE limited to this use case.

Variants of gradient-ascent can be used to address *pulse parametrizations*. That is, the control parameters may be arbitrary parameters of the control field (e.g., spectral coefficients) instead of the field amplitude $\epsilon_n$ in a particular time interval. This is often relevant to design control fields that meet experimental constraints. One possible realization is to calculate the gradients for the control parameters from the gradients of the time-discrete control amplitudes via the chain rule [86–89]. This approach has recently been named "GRadient Optimization Using Parametrization" (GROUP) [90]. An implementation of several variants of GROUP is available in the QEngine C++ library [91]. An alternative for a moderate number of control parameters is "gradient-optimization of analytic controls" (GOAT) [92]. GOAT evaluates the relevant gradient with forward-mode differentiation; that is, $\partial \tau_k / \partial \epsilon_n$ is directly evaluated alongside $\tau_k$. For $N = |\{\epsilon_m\}|$ control parameters, this implies $N$ forward propagations of the state-gradient pair per iteration. Alternatively, the $N$ propagations can be concatenated into a single propagation in a Hilbert space enlarged by a factor $N$ (the original state paired with $N$ gradients).

A benefit of GOAT over the more general GROUP is that it does not piggy-back on the piecewise-constant discretization of the control field, and thus may avoid the associated numerical error. This allows to optimize to extremely high fidelities as required for some error correction protocols [92].

## 4.4 Gradient-free optimization

In situations where the problem can be reduced to a relatively small number of control parameters (typically less than $\approx 20$, although this number may be pushed to $\approx 50$ by sequential increase of the number of parameters and re-parametrization [93, 94]), gradient-free optimization becomes feasible. The most straightforward use case are controls with an analytic

shape (e.g. due to the constraints of an experimental setup), with just a few free parameters. As an example, consider control pulses that are restricted to a Gaussian shape, so that the only free parameters are peak amplitude, pulse width and delay. The control parameters are not required to be parameters of a time-dependent control, but may also be static parameters in the Hamiltonian, e.g. the polarization of the laser beams utilized in an experiment [95].

A special case of gradient-free optimization is the Chopped RAndom Basis (CRAB) method [96,97]. The essence of CRAB is in the specific choice of the parametrization in terms of a low-dimensional *random* basis, as the name implies. Thus, it can be used when the parametrization is not pre-defined as in the case of direct free parameters in the pulse shape discussed above. The optimization itself is normally performed by Nelder-Mead simplex based on this parametrization, although any other gradient-free method could be used as well. An implementation of CRAB is available in QuTiP. CRAB is prone to getting stuck in local minima of the optimization landscape. To remedy this, a variant of CRAB, "dressed CRAB" (DCRAB) has been developed [93] that re-parametrizes the controls when this happens.

Gradient-free optimization does not require backward propagation, only forward propagation of the initial states and evaluation of the optimization functional $J$. The functional is not required to be analytic. It may be of a form that does not allow calculation of the gradients $\partial J_T / \partial \langle \phi_k |$ (Krotov's method) or $\partial J / \partial \epsilon_j$ (GRAPE). The optimization also does not require any storage of states. However, the number of iterations can grow extremely large, especially with an increasing number of control parameters. Thus, an optimization with a gradient-free method is not necessarily more efficient overall compared to a gradient-based optimization with much faster convergence. For only a few parameters, however, it can be highly efficient. This makes gradient-free optimization useful for "pre-optimization", that is, for finding guess controls that are then further optimized with a gradient-based method [35].

Generally, gradient-free optimization can be easily realized directly in QuTiP or any other software package for the simulation of quantum dynamics:

- Write a function that takes an array of optimization parameters as input and returns a figure of merit. This function would, e.g., construct a numerical control pulse from the control parameters, simulate the dynamics using `qutip.mesolve.mesolve`, and evaluate a figure of merit (like the overlap with a target state).

- Pass the function to `scipy.optimize.minimize` for gradient-free optimization.

The implementation in `scipy.optimize.minimize` allows to choose between different optimization methods, with Nelder-Mead simplex being the default. There exist also more advanced optimization methods available in packages like NLopt [98] or Nevergrad [99] that may be worth exploring for improvements in numerical efficiency and additional functionality such as support for non-linear constraints.

## 4.5 Choosing an optimization method

In the following, we discuss some of the concerns in the choice of optimization methods. The discussion is limited to iterative open-loop methods, where the optimization is based on a numerical simulation of the dynamics. It excludes analytical control methods such as geometric control, closed-loop methods, or coherent feedback control; see Ref. [100] for an overview.

Whether to use a gradient-free optimization method, GRAPE, or Krotov's method depends on the size of the problem, the requirements on the control fields, and the mathematical properties of the optimization functional. Gradient-free methods should be used if the number of independent control parameters is smaller than $\approx 20$, or the functional is of a form that does not allow to calculate gradients easily. It is always a good idea to use a gradient-free method to obtain improved guess pulses for use with a gradient-based method [35].

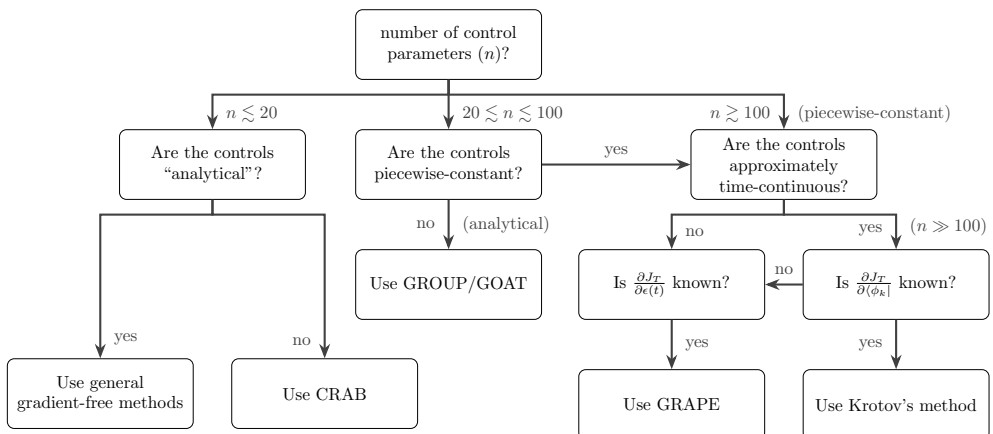

Figure 2: Decision tree for the choice of a numerical open-loop optimization method. The choice of control method is most directly associated with the number of control parameters ($n$). For "piecewise-constant controls", the control parameters are the values of the control field in each time interval. For "analytical" controls, we assume that the control fields are described by a fixed analytical formula parametrized by the control parameters. The "non-analytical" controls for CRAB refer to the *random* choice of a fixed number of spectral components, where the control parameters are the coefficients for those spectral components. Each method in the diagram is meant to include all its variants, a multitude of gradient-free methods and e.g. DCRAB for CRAB, GRAPE-LBFGS and sequential/hybrid gradient-descent for GRAPE, and K-BFGS for Krotov's method, see text for detail.

GRAPE or its variants should be used if the control parameters are discrete, such as on a coarse-grained time grid, and the derivative of $J$ with respect to each control parameter is easily computable. Note that the implementation provided in QuTiP is limited to state-to-state transitions and quantum gates, even though the method is generally applicable to a wider range of objectives.

When the control parameters are general analytic coefficients instead of time-discrete amplitudes, the GROUP [87, 88, 90] or GOAT [92] variant of gradient-ascent may be a suitable choice. GOAT in particular can avoid the numerical error associated with time discretization. However, as the method scales linearly in memory and/or CPU with the number of control parameters, this is best used when then number of parameters is below 100.

Krotov's method should be used if the control is close to time-continuous, and if the derivative of $J_T$ with respect to the states, Eq. (12), can be calculated. When these conditions are met, Krotov's method gives excellent convergence. The general family of monotonically convergent iteration schemes [64] may also be used.

The decision tree in Fig. 2 can guide the choice of an optimization method. The key deciding factors are the number of control parameters ($n$) and whether the controls are time-discrete. Of course, the parametrization of the controls is itself a choice. Sometimes, experimental constraints only allow controls that depend on a small number of tunable parameters. However, this necessarily limits the exploration of the full physical optimization landscape. At the other end of the spectrum, arbitrary time-continuous controls such as those assumed in Krotov's method have no inherent constraints and are especially useful for more fundamental tasks, such as mapping the design landscape of a particular system [101] or determining the quantum speed limit, i.e., the minimum time in which the system can reach a given target [15, 102, 103].

## 5  Future perspectives

While the present implementation of the `krotov` Python package already provides the user with the capability to tackle a broad range of optimization targets in quantum optimal control, possible future additions could enhance its versatility even further. A first most welcome extension concerns the capability to parametrize the pulse. This would allow to guarantee positivity of the control field when optimizing, e.g., Rabi frequencies instead of pulse amplitudes, or provide a straightforward way to impose an upper bound $\epsilon_0$ on the field amplitude. The latter could be achieved, for example, by way of defining $\epsilon(t) = \epsilon_0 \tanh^2(u(t))$ [104]. The simplest approach to adapt the algorithm to such parametrizations is to consider the Hamiltonian / Liouvillian as a function of $u(t)$ instead of $\epsilon(t)$. Then, the update equation will also be formulated with respect to $u(t)$ and once the optimization is completed the physical pulse $\epsilon(t)$ can be obtained by direct evaluation. A caveat in this approach is the fact that the Hamiltonian / Liouvillian will not be a linear function of $u(t)$ even if it was linear with respect to $\epsilon(t)$. As such, additional care needs to be taken regarding the choice of a sufficiently large value for the inverse step size $\lambda_a$ to preserve monotonic convergence [22].

A second feature worthwhile to add in a future version of the `krotov` Python package are state-dependent constraints $g_b \neq 0$ [22,53]. This would enable to optimization towards time-dependent targets [51,52]. If the constraint is a non-convex function of the states, usage of the second-order contribution, $\sigma(t) \neq 0$, in the Krotov update equation (31) is required to ensure monotonic convergence. In this case, $\sigma(t) \neq 0$ is linearly time-dependent [22]. The presence of a state-dependent constraint also implies a source term in the equation of motion for the adjoint states, cf. Eq. (30). Although this source term may pose some numerical challenges for differential equation solvers, it should be noted that the solution of a linear Schrödinger equation with a source term already allows for solving Schrödinger equations with a general nonlinearity [105]. Assuming an appropriate solver was available, the `krotov` package would have to calculate the appropriate source term and pass it to that solver.

Finally, the current implementation of the package does not yet allow for imposing spectral constraints in the optimization functional, although this is in principle possible in Krotov's method [106,107]. At first glance, it may be surprising that a method that updates the control sequentially (time-locally) can include spectral (time-global) constraints without breaking monotonic convergence. The key insight is to generalize $g_a(\epsilon(t))$, Eq. (7), to a time-non-local form,

$$g_a(\epsilon(t), t) = \int_0^T \Delta\epsilon(t) K(t - t') \Delta\epsilon(t') \, dt' . \tag{26}$$

Provided the kernel $K(\tau)$ encoding the spectral constraint via a Fourier transform is positive semi-definite, Krotov's method yields a monotonically converging optimization algorithm [107]. However, the price to pay is the need to solve a Fredholm equation of the second kind, which has not yet been implemented numerically. It should be noted that the current version of the `krotov` package already supports a less rigorous method to limit the spectral width of the optimized controls, by applying a simple spectral filter after each iteration. By mixing the unfiltered and filtered controls, monotonic convergence can be preserved [108].

The above mentioned features concern direct extensions of Krotov's method that have already been reported in the literature. Beyond that, Krotov's method could also be combined with other optimization approaches to overcome some of its inherent limitations. The most severe limitations are that Krotov's method requires analytically computable derivatives, see Eq. (12), and it searches only in the local region of the point in the optimization landscape where that derivative is being evaluated (as any gradient-based method does). The optimized pulse thus depends on the guess pulse from which the optimization starts. If the pulse can be parametrized with only a few relevant parameters, the search can be effectively globalized by

scanning those parameters [36]. This approach becomes more efficient when pre-optimizing the parameters with a gradient-free method [35]. In this respect, it will be worthwhile to combine the krotov package with the nonlinear optimization toolbox NLopt [98] containing several global search methods. This should not only improve the convergence of the pre-optimization compared to using the simplex method [35] but would, moreover, also allow for simultaneously optimizing time-dependent and time-independent controls. The inherent limitation of requiring computable derivatives might be lifted by combining Krotov's method with automatic differentiation, similar to what has been achieved for gradient-based optimization in the spirit of GRAPE [84, 85]. Finally, it would also be interesting to analyze optimizations using Krotov's method with machine learning techniques [109].

## 6 Conclusions

We have presented the Python implementation of Krotov's method for quantum optimal control that comes with a number of example use cases, suitable in particular for applications in quantum information science. The hallmark of Krotov's method is fast initial convergence, monotonicity and aptitude for time-continuous controls.

The krotov package adds to the available tools for optimal control around the popular Quantum Toolbox in Python (QuTiP). The QuTiP package itself contains routines for gradient-free optimization and gradient-ascent, currently limited to state-to-state transitions or quantum gates and to a coarse time grid. Our package provides an interface for formulating quantum control problems that lifts these limitations and aims to be sufficiently general to describe *any* problem in quantum control. In future work, the same interface may be used to drive optimization methods beyond Krotov's method, enabling direct comparison of different methods.

We have given an overview of the most important gradient-free and gradient-based methods that have been developed thus far. Each method has its own strengths and weaknesses under different constraints. Krotov's method in particular excels at finding the *least* constrained control fields and is thus particularly useful for exploring the fundamental limits of control in a given quantum system. On the other hand, when there are in fact strong external constraints on the controls due to experimental limitations, other methods may have an advantage. Our discussion will allow the reader to make an informed choice for the most appropriate method.

Our implementation of Krotov's method together with the examples and explanations in this paper, and the pseudocode in Appendix B may serve as a reference when implementing Krotov's method in other systems or languages. We hope that this will motivate wider adoption of Krotov's method, and the use of optimal quantum control in general. As quantum technology matures, optimal control for solving the inherently difficult design problems will only gain in importance. Thus, the creation of a high quality open source software stack around optimal control is paramount. The krotov package is a contribution to this endeavor.

## Acknowledgements

M.H.G was supported by the Army Research Laboratory under Cooperative Agreement Number W911NF-17-2-0147. The Kassel team gratefully acknowledges financial support from the Volkswagenstiftung, the European Union's Horizon 2020 research and innovation programme under the Marie Sklodowska-Curie grant agreement Nr. 765267, and the State Hessen Initiative for the Development of Scientific and Economic Excellence (LOEWE) within the focus project SMolBits. We thank Steffen Glaser, Shai Machnes, and Nathan Shammah for fruitful discussions and comments.

# A The Krotov update equation

The core of Krotov's method is the numerical evaluation of the field update in each iteration, $\Delta\epsilon_l^{(i)}(t)$ in Eq. (9). In the following, we specify $\Delta\epsilon_l^{(i)}(t)$ and discuss how its discretization leads to a numerical scheme.

## A.1 First order update

Krotov's method is based on a rigorous examination of the conditions for calculating the updated fields $\{\epsilon_l^{(i)}(t)\}$ such that $J(\{|\phi_k^{(i)}(t)\rangle\}, \{\epsilon_l^{(i)}(t)\}) \leq J(\{|\phi_k^{(i-1)}(t)\rangle\}, \{\epsilon_l^{(i-1)}(t)\})$ is true *by construction* [22, 45, 46, 49, 50]. For a general functional of the form in Eq. (4), with a convex final-time functional $J_T$, the condition for monotonic convergence is

$$\frac{\partial g_a}{\partial \epsilon_l(t)}\bigg|_{(i)} = 2\,\text{Im}\left[\sum_{k=1}^{N}\left\langle\chi_k^{(i-1)}(t)\left|\left(\frac{\partial\hat{H}}{\partial\epsilon_l(t)}\bigg|_{(i)}\right)\right|\phi_k^{(i)}(t)\right\rangle\right], \tag{27}$$

see Ref. [50]. The notation for the derivative on the right hand side being evaluated at $(i)$ should be understood to apply when the control Hamiltonian is not linear so that $\frac{\partial\hat{H}}{\partial\epsilon_l(t)}$ is still time-dependent; the derivative must then be evaluated for $\epsilon_l^{(i)}(t)$ – or, numerically, for $\epsilon_l^{(i-1)}(t) \approx \epsilon_l^{(i)}(t)$. If there are multiple controls, Eq. (27) holds for every control field $\epsilon_l(t)$ independently.

For $g_a$ as in Eq. (7), this results in an *update* equation [5, 49, 50],

$$\Delta\epsilon_l^{(i)}(t) = \frac{S_l(t)}{\lambda_{a,l}}\,\text{Im}\left[\sum_{k=1}^{N}\left\langle\chi_k^{(i-1)}(t)\left|\left(\frac{\partial\hat{H}}{\partial\epsilon_l(t)}\bigg|_{(i)}\right)\right|\phi_k^{(i)}(t)\right\rangle\right], \tag{28}$$

cf. Eq. (22), with the equation of motion for the forward propagation of $|\phi_k^{(i)}\rangle$ under the optimized controls $\{\epsilon_l^{(i)}(t)\}$ of the iteration $(i)$,

$$\frac{\partial}{\partial t}\left|\phi_k^{(i)}(t)\right\rangle = -\frac{i}{\hbar}\hat{H}^{(i)}\left|\phi_k^{(i)}(t)\right\rangle. \tag{29}$$

The co-states $|\chi_k^{(i-1)}(t)\rangle$ are propagated backwards in time under the guess controls of iteration $(i)$, i.e., the optimized controls from the previous iteration $(i-1)$, as

$$\frac{\partial}{\partial t}\left|\chi_k^{(i-1)}(t)\right\rangle = -\frac{i}{\hbar}\hat{H}^{\dagger(i-1)}\left|\chi_k^{(i-1)}(t)\right\rangle + \frac{\partial g_b}{\partial\langle\phi_k|}\bigg|_{(i-1)}, \tag{30}$$

with the boundary condition of Eq. (12).

The coupled equations (28–30) can be generalized to open system dynamics by replacing Hilbert space states with density matrices, $\hat{H}$ with $i\mathcal{L}$, and brakets with Hilbert-Schmidt products, $\langle\cdot|\cdot\rangle \rightarrow \langle\langle\cdot|\cdot\rangle\rangle$. In full generality, $\hat{H}$ in Eq. (28) is the operator $H$ on the right-hand side of whatever the equation of motion for the forward propagation of the states is, written in the form $i\hbar\dot{\phi} = H\phi$, cf. Eq. (29). Note also that the backward propagation Eq. (30) uses the adjoint $H$, which is relevant both for a dissipative Liouvillian [30, 110, 111] and a non-Hermitian Hamiltonian [28, 112].

## A.2 Second order update

The update Eq. (28) assumes that the equation of motion is linear ($\hat{H}$ does not depend on the states $|\phi_k(t)\rangle$), the functional $J_T$ is convex, and no state-dependent constraints are used

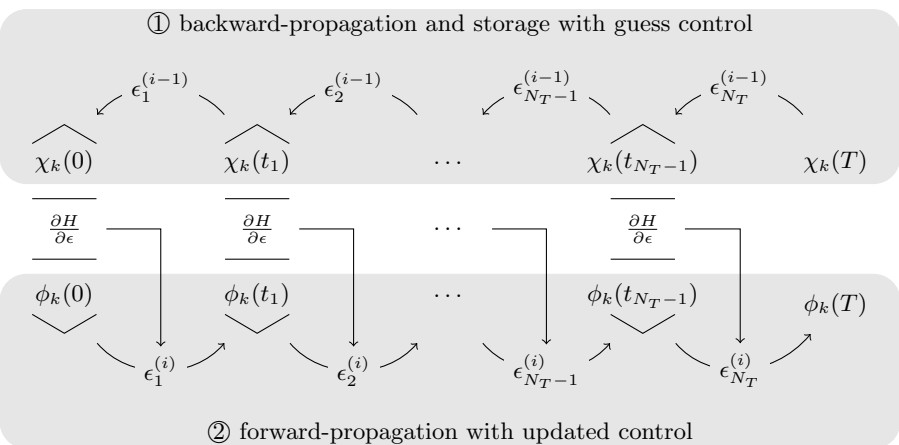

① backward-propagation and storage with guess control

② forward-propagation with updated control

Figure 3: Sequential update scheme in Krotov's method on a time grid.

($g_b \equiv 0$). When any of these conditions are not fulfilled, it is still possible to derive an optimization algorithm with monotonic convergence via a "second order" term in Eqs. (27, 28) [22,46], The full update equation then reads

$$\Delta\epsilon_l^{(i)}(t) = \frac{S_l(t)}{\lambda_{a,l}} \operatorname{Im}\left[ \sum_{k=1}^{N}\left\langle \chi_k^{(i-1)}(t)\left|\left(\left.\frac{\partial\hat{H}}{\partial\epsilon_l(t)}\right|_{(i)}\right)\right|\phi_k^{(i)}(t)\right\rangle \right.$$
$$\left. +\frac{1}{2}\sigma(t)\left\langle \Delta\phi_k^{(i)}(t)\left|\left(\left.\frac{\partial\hat{H}}{\partial\epsilon_l(t)}\right|_{(i)}\right)\right|\phi_k^{(i)}(t)\right\rangle \right], \tag{31}$$

with

$$|\Delta\phi_k^{(i)}(t)\rangle \equiv |\phi_k^{(i)}(t)\rangle - |\phi_k^{(i-1)}(t)\rangle, \tag{32}$$

see Ref. [22] for the full construction of the second-order condition.

In Eq. (31), $\sigma(t)$ is a scalar function that must be properly chosen to ensure monotonic convergence. As shown in Ref. [22], it is possible to numerically approximate $\sigma(t)$, see Section 3.4 for an example.

## A.3 Time discretization

The derivation of Krotov's method assumes time-continuous control fields. Only in this case, monotonic convergence is mathematically guaranteed. However, for practical numerical applications, we have to consider controls on a discrete time grid with $N_T+1$ points running from $t = t_0 = 0$ to $t = t_{N_T} = T$, with a time step $dt$. The states are defined on the points of the time grid, while the controls are assumed to be constant on the intervals of the time grid. A coarse time step must be compensated by larger values of the inverse step size $\lambda_{a,l}$, slowing down convergence. Values that are too small will cause sharp spikes in the optimized control and numerical instabilities. A lower limit for $\lambda_{a,l}$ can be determined from the requirement that the change $\Delta\epsilon^{(i)}(t)$ should be at most of the same order of magnitude as the guess pulse $\epsilon^{(i-1)}(t)$ for that iteration. The Cauchy-Schwarz inequality applied to the update equation (28) yields

$$\lambda_a \geq \frac{1}{\left\|\epsilon^{(i-1)}(t)\right\|_\infty}\left[\sum_k\left\|\left|\chi_k^{(i-1)}(t)\right\rangle\right\|_\infty\right]\left\|\frac{\partial\hat{H}}{\partial\epsilon(t)}\right\|_\infty. \tag{33}$$

From a practical point of view, the best strategy is to start the optimization with a comparatively large value of $\lambda_{a,l}$, and after a few iterations lower $\lambda_{a,l}$ as far as possible without introducing

numerical instabilities. The value of $\lambda_{a,l}$ may be adjusted dynamically with respect to the rate of convergence, via the `modify_params_after_iter` argument to the `optimize_pulses` function. Generally, the ideal choice of $\lambda_{a,l}$ requires some trial and error.

The discretization yields the numerical scheme shown in Fig. 3 for a single control field (no index $l$), and assuming the first-order update is sufficient to guarantee monotonic convergence for the chosen functional. For simplicity, we also assume that the Hamiltonian is linear in the control, so that $\partial \hat{H} / \partial \epsilon(t)$ is not time-dependent. The scheme proceeds as follows:

1. Construct the states $\{|\chi_k^{(i-1)}(T)\rangle\}$ according to Eq. (12). For most functionals, specifically any that are more than linear in the overlaps $\tau_k$ defined in Eq. (6), the states $\{|\chi_k^{(i-1)}(T)\rangle\}$ depend on the states $\{|\phi_k^{(i-1)}(T)\rangle\}$ forward-propagated under the optimized pulse from the previous iteration, that is, the guess pulse in the current iteration.

2. Perform a backward propagation using Eq. (30) as the equation of motion over the entire time grid. The resulting state at each point in the time grid must be stored in memory.

3. Starting from the known initial states $\{|\phi_k\rangle\} = \{|\phi_k(t = t_0 = 0)\rangle\}$, calculate the pulse update for the first time step according to

$$\Delta\epsilon_1^{(i)} \equiv \Delta\epsilon^{(i)}(\tilde{t}_0) = \frac{S(\tilde{t}_0)}{\lambda_a} \mathrm{Im}\left[\sum_{k=1}^{N}\left\langle \chi_k^{(i-1)}(t_0) \left| \frac{\partial \hat{H}}{\partial \epsilon} \right| \phi_k(t_0) \right\rangle\right]. \tag{34}$$

The value $\Delta\epsilon_1^{(i)}$ is taken on the midpoint of the first time interval, $\tilde{t}_0 \equiv t_0 + \mathrm{d}t/2$, based on the assumption of a piecewise-constant control field and an equidistant time grid with spacing $\mathrm{d}t$.

4. Use the updated field $\epsilon_1^{(i)}$ for the first interval to propagate $|\phi_k(t = t_0)\rangle$ for a single time step to $|\phi_k^{(i)}(t = t_0 + \mathrm{d}t)\rangle$, with Eq. (29) as the equation of motion. The updates then proceed sequentially, using the discretized update equation

$$\Delta\epsilon_{n+1}^{(i)} \equiv \Delta\epsilon^{(i)}(\tilde{t}_n) = \frac{S(\tilde{t}_n)}{\lambda_a} \mathrm{Im}\left[\sum_{k=1}^{N}\left\langle \chi_k^{(i-1)}(t_n) \left| \frac{\partial \hat{H}}{\partial \epsilon} \right| \phi_k^{(i)}(t_n) \right\rangle\right], \tag{35}$$

with $\tilde{t}_n \equiv t_n + \mathrm{d}t/2$ for each time interval $n$, until the final forward-propagated state $|\phi_k^{(i)}(T)\rangle$ is reached.

5. The updated control field becomes the guess control for the next iteration of the algorithm, starting again at step 1. The optimization continues until the value of the functional $J_T$ falls below some predefined threshold, or convergence is reached, i.e., $\Delta J_T$ approaches zero so that no further significant improvement of $J_T$ is to be expected.

Eq. (28) re-emerges as the continuous limit of the time-discretized update equation (35), i.e., $\mathrm{d}t \to 0$ so that $\tilde{t}_n \to t_n$. Note that Eq. (35) resolves the seeming contradiction in the time-continuous Eq. (28) that the calculation of $\epsilon^{(i)}(t)$ requires knowledge of the states $|\phi_k^{(i)}(t)\rangle$ which would have to be obtained from a propagation under $\epsilon^{(i)}(t)$. By having the time argument $\tilde{t}_n$ on the left-hand-side of Eq. (35), and $t_n < \tilde{t}_n$ on the right-hand-side (with $S(\tilde{t}_n)$ known at all times), the update for each interval only depends on "past" information.

# B   Pseudocode for Krotov's method

For reference, Algorithm 1 shows the complete pseudocode of an optimization with Krotov's method, as implemented in the `krotov` package. It realizes the time-discretized scheme described in Appendix A.3.

Variables are color coded. Scalars are set in blue, e.g. $\epsilon_{ln}^{(0)}$. States (Hilbert space states or vectorized density matrices) are set in purple, e.g. $\phi_k^{\text{init}}$. They may be annotated with light gray superscripts to indicate the iteration-index $i$ of the control under which state was propagated, and with light gray time arguments. These annotations serve only to connect the variables to the equations in Appendix A: $\phi_k^{(0)}(t_n)$ and $\phi_k^{(0)}(t_{n-1})$ are the same variable $\phi_k$. Operators acting on states are set in green, e.g. $\mu_{lkn}$. These may be implemented as a sparse matrix or implicitly as a function that returns the result of applying the operator to a state. Lastly, storage arrays are set in red, e.g. $\Phi_0$. Each element of a storage array is a state.

The Python implementation groups several of the algorithm's input parameters by introducing a list of $N$ "objectives". The objectives are indexed by $k$, and each objective contains the initial state $\phi_k^{\text{init}}$, the Hamiltonian or Liouvillian $H_k$ to be used by the propagator $U$ and for the operators $\mu_{lkn}$, and possibly a "target" to be taken into account by the function $\chi$. In many applications, $H_k \equiv H$ is the same in all objectives, and $\mu_{lkn} \equiv \mu_l$ if $H$ is linear in the controls in addition. The subscript $n$ and the superscript $(i-1)$ for $\mu_{lkn}^{(i-1)}$ in lines 31, 34 comes into play only if $H$ is *not* linear in the control. Mathematically, $\mu_{lkn}$ would then have to be evaluated using the *updated* control. Since the update is not yet known, the *guess* control may be used as an approximation (valid for sufficiently large $\lambda_{a,l}$).

The CPU resources required for the optimization are dominated by the time propagation (calls to the function $U$ in lines 7, 24 37). This is under the assumption that evaluating $U$ dominates the application of the operator $\mu_{lkn}^{(i-1)}$ to the state $\phi_k^{(i)}(t_{n-1})$ and the evaluation of the inner product of two states, lines 31, 34. This condition is fulfilled for any non-trivial Hilbert space dimension.

Loops over the index $k$ are parallelizable, in particular in a shared-memory (multi-threaded) parallelization environment like OpenMP. In a (multi-process) method-passing environment like MPI, some care must be taken to minimize communication overhead from passing large state vectors. For some (but not all) functionals, inter-process communication can be reduced to only the scalar values constituting the sum over $k$ in lines 31, 34.

The memory requirements of the algorithm are dominated by the storage arrays $\Phi_0$, $\Phi_1$, and $X$. Each of these must store $N(N_T + 1)$ full state vectors (a full time propagation for each of the $N$ objectives). Each state vector is typically an array of double-precision complex numbers. For a Hilbert space dimension $d$, a state vector thus requires $16d$ bytes of memory, or $16d^2$ bytes for a density matrix. Under certain conditions, the use of $\Phi_0$ and $\Phi_1$ can be avoided: both are required only when the second order update is used ($\sigma(t) \neq 0$). When the first order update is sufficient, $\Phi_1$ may overwrite $\Phi_0$ so that the two collapse into a single forward-storage $\Phi$. The states stored in $\Phi$ are only used for the inhomogeneity $\partial g_b / \partial \langle \phi_k |$ in Eq. (30), and no storage $\Phi$ of forward-propagated states at all is required if $g_b \equiv 0$. Thus, in most examples, only the storage $X$ of the backward-propagated states remains. In principle, if the time propagation $U$ is unitary (i.e., invertible), the states stored in $X$ could be recovered by forward-propagation of $\{\chi_k^{(i-1)}(t=0)\}$, eliminating $X$ at the (considerable) runtime cost of an additional time propagation.

---

**Algorithm 1** KROTOV'S METHOD FOR QUANTUM OPTIMAL CONTROL

**Input:**

1. list of guess control values $\{\epsilon_{ln}^{(0)}\}$ where $\epsilon_{ln}^{(0)}$ is the value of the $l$'th control field on the $n$'th interval of the propagation time grid ($t_0 = 0, \ldots, t_{N_T} = T$), i.e., $\epsilon_{ln}^{(0)} \equiv \epsilon_l^{(0)}(\tilde{t}_{n-1})$ with $n \in [1, N_T]$ and $\tilde{t}_n \equiv (t_n + t_{n+1})/2$

2. list of update-shape values $\{S_{ln}\}$ with each $S_{ln} \in [0, 1]$

3. list of update step size values $\{\lambda_{a,l}\}$

4. list of $N$ initial states $\{\phi_k^{\text{init}}\}$ at $t = t_0 = 0$

5. propagator function $U$ that in "forward mode" receives a state $\phi_k(t_n)$ and a list of control values $\{\epsilon_{ln}\}$ and returns $\phi_k(t_{n+1})$ by solving the differential equation (29), respectively in "backward mode" (indicated as $U^\dagger$) receives a state $\chi_k(t_n)$ and returns $\chi_k(t_{n-1})$ by solving the differential equation (30)

6. list of operators $\mu_{lkn} = \frac{\partial H_k}{\partial \epsilon_l(t)}\big|_{\epsilon_{ln}}$, cf. Eq. (27), where $H_k$ is the right-hand-side of the equation of motion of $\phi_k(t)$, up to a factor of $(-i/\hbar)$, cf. Eq. (29)

7. function $\chi$ that receives a list of states $\{\phi_k(T)\}$ and returns a list of states $\{\chi_k(T)\}$ according to Eq. (12)

8. optionally, if a second order construction of the pulse update is necessary: function $\sigma(t)$

**Output:** optimized control values $\{\epsilon_{ln}^{(\text{opt})}\}$, such that $J[\{\epsilon_{ln}^{(\text{opt})}\}] \leq J[\{\epsilon_{ln}^{(0)}\}]$, with $J$ defined in Eq. (4).

1: **procedure** KROTOVOPTIMIZATION($\{\epsilon_{ln}^{(0)}\}, \{S_{ln}\}, \{\lambda_{a,l}\}, \{\phi_k^{\text{init}}\}, U, \{\mu_{lkn}\}, \chi, \sigma$ )
2:      $i \leftarrow 0$                                                 ▷ iteration number
3:      allocate forward storage array $\Phi_0[1 \ldots N, 0 \ldots N_T]$
4:      **for** $k \leftarrow 1, \ldots, N$ **do**                          ▷ initial forward-propagation
5:          $\Phi_0[k, 0] \leftarrow \phi_k^{(0)}(t_0) \leftarrow \phi_k^{\text{init}}$
6:          **for** $n \leftarrow 1, \ldots, N_T$ **do**
7:              $\Phi_0[k, n] \leftarrow \phi_k^{(0)}(t_n) \leftarrow U(\phi_k^{(0)}(t_{n-1}), \{\epsilon_{ln}^{(0)}\})$          ▷ propagate and store
8:          **end for**
9:      **end for**
10:      **while** not converged **do**                               ▷ optimization loop
11:          $i \leftarrow i + 1$
12:          $\Phi_1, \{\epsilon_{ln}^{(i)}\} \leftarrow$ KROTOVITERATION($\Phi_0, \{\epsilon_{ln}^{(i-1)}\}, \ldots$)
13:          $\Phi_0 \leftarrow \Phi_1$
14:      **end while**
15:      $\forall l, \forall n : \epsilon_{ln}^{(\text{opt})} \leftarrow \epsilon_{ln}^{(i)}$                              ▷ final optimized controls
16: **end procedure**

**Notes:**

- The index $k$ numbers the independent states to be propagated, respectively the independent "objectives" (see text for details), $l$ numbers the independent control fields, and $n$ numbers the intervals on the time grid. All of these indices start at 1.

- The optimization loop may be stopped if the optimization functional or the change of functional falls below a pre-defined threshold, a maximum number of iterations is reached, or any other criterion.

---

17: **procedure** KROTOVITERATION($\Phi_0$, $\{\epsilon_{ln}^{(i-1)}\}$, $\{S_{ln}\}$, $\{\lambda_{a,l}\}$, $\{\phi_k^{\text{init}}\}$, $U$, $\{\mu_{lkn}\}$, $\chi$, $\sigma$ )

18:     $\forall k : \phi_k^{(i-1)}(T) \leftarrow \Phi_0[k, N_T]$

19:     $\{\chi_k^{(i-1)}(T)\} \leftarrow \chi(\{\phi_k^{(i-1)}(T)\})$     ▷ backward boundary condition

20:     allocate backward storage array $X[1\ldots N, 0\ldots N_T]$.

21:     **for** $k \leftarrow 1, \ldots, N$ **do**

22:         $X[k, N_T] \leftarrow \chi_k^{(i-1)}(T)$

23:         **for** $n \leftarrow N_T, \ldots, 1$ **do**     ▷ backward-propagate and store

24:             $X[k, n-1] \leftarrow \chi_k^{(i-1)}(t_{n-1}) \leftarrow U^\dagger(\chi_k^{(i-1)}(t_n), \{\epsilon_{ln}^{(i-1)}\}, \Phi_0)$

25:         **end for**

26:     **end for**

27:     allocate forward storage array $\Phi_1[1\ldots N, 0\ldots N_T]$

28:     $\forall k : \Phi_1[k, 0] \leftarrow \phi_k^{(i)}(t_0) \leftarrow \phi_k^{\text{init}}$

29:     **for** $n \leftarrow 1, \ldots, N_T$ **do**     ▷ sequential update loop

30:         $\forall k : \chi_k^{(i-1)}(t_{n-1}) \leftarrow X[k, n-1]$

31:         $\forall l : \Delta\epsilon_{ln} \leftarrow \frac{S_{ln}}{\lambda_{a,l}} \text{Im} \sum_k \langle \chi_k^{(i-1)}(t_{n-1}) | \mu_{lkn}^{(i-1)} | \phi_k^{(i)}(t_{n-1}) \rangle$     ▷ first order

32:         **if** $\sigma(t) \neq 0$ **then**     ▷ second order

33:             $\forall k : \Delta\phi_k^{(i)}(t_{n-1}) \leftarrow \phi_k^{(i)}(t_{n-1}) - \Phi_0[k, n-1]$

34:             $\forall l : \Delta\epsilon_{ln} \leftarrow \Delta\epsilon_{ln} + \frac{S_{l,n-1}}{\lambda_{a,l}} \text{Im} \sum_k \frac{1}{2}\sigma(\tilde{t}_n) \langle \Delta\phi_k^{(i)}(t_{n-1}) | \mu_{lkn}^{(i-1)} | \phi_k^{(i)}(t_{n-1}) \rangle$

35:         **end if**

36:         $\forall l : \epsilon_{ln}^{(i)} \leftarrow \epsilon_{ln}^{(i-1)} + \Delta\epsilon_{ln}$     ▷ apply update

37:         $\forall k : \Phi_1[k, n] \leftarrow \phi_k^{(i)}(t_n) \leftarrow U(\phi_k^{(i)}(t_{n-1}), \{\epsilon_{ln}^{(i)}\})$     ▷ propagate and store

38:     **end for**

39:     **if** $\sigma(t) \neq 0$ **then**

40:         Update internal parameters of $\sigma(t)$ if necessary, using $\Phi_0$, $\Phi_1$

41:     **end if**

42: **end procedure**

**Notes:**

- The braket notation in line 31 indicates the (Hilbert-Schmidt) inner product of the state $\chi_k^{(i-1)}(t_n-1)$ and the state resulting from applying $\mu_{lkn}^{(i-1)}$ to $\phi_k^{(i)}(t_{n-1})$. In Hilbert space, this is the standard braket. In Liouville space, it is $\text{tr}\left(\chi_k^{\dagger} \mu_{lkn}[\phi_k]\right)$ with density matrices $\chi_k$, $\phi_k$ and a super-operator $\mu_{lkn}$.

- For numerical stability, the states $\chi_k^{(i-1)}(T)$ in line 19 may be normalized. This norm then has to taken into account in the pulse update, line 31.

- In line 24, the storage array $\Phi_0$ is passed to $U^\dagger$ only to account for the inhomogeneity due to a possible state-dependent constraint, $\partial g_b / \partial \langle \phi_k |$ in Eq. (30). If $g_b \equiv 0$, the parameter can be omitted.

## C   Installation instructions

The `krotov` package is available for Python versions ≥3.5. Its main dependency is QuTiP [24, 25]. Thus, you should consider QuTiP's installation instructions, see http://qutip.org.

It is strongly recommended to install Python packages into an isolated environment. One possible system for creating such environments it `conda`, available as part of the Anaconda Python Distribution, respectively the smaller "Miniconda", available at https://conda.io/miniconda.html. Anaconda has the additional benefit that it provides bi-

nary versions of scientific Python packages that include compiled extensions, and may be hard to install on systems that lack the necessary compilers (Windows, macOS). This includes the QuTiP package. Assuming `conda` is installed, the following commands set up a virtual environment into which the `krotov` package can then be installed:

```
$ conda create -n qucontrolenv python=3.7
$ conda activate qucontrolenv
$ conda config --append channels conda-forge
$ conda install qutip
```

To install the latest released version of `krotov` into your current (conda) environment, run this command in your terminal:

```
$ pip install krotov
```

The examples in the online documentation and in Section 2.4 require additional dependencies. These can be installed with

```
$ pip install krotov[dev]
```

See the package documentation linked in Appendix D for the most current installation instructions.

## D  Package documentation

This paper describes only the most central features of the `krotov` package. For a complete documentation, refer to https://qucontrol.github.io/krotov. The most current version of the `krotov` package is available at https://github.com/qucontrol/krotov under a BSD license.

The example script of Section 2.4 is available at https://github.com/qucontrol/krotov/tree/paper/examples. A Jupyter notebook version of the same example is available in the Examples section of the online documentation, together with notebooks illustrating in more detail the optimization tasks discussed in Section 3.

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
