# Peer review of "Krotov: A Python implementation of Krotov's method for quantum optimal control"

_SciPost Physics, doi:SciPost Phys. 7, 080 (2019)_

## Round 3 · Referee Report · Anonymous (Referee 1) · 2019-5-20

Strengths

  1. The Krotov method is well-known in the optimal control community.

  2. The addition of a high-quality open-source implementation is very welcome and could be of use to a large audience of both optimal control theoreticians and experimentalists.

Weaknesses

  1. The description of the base Krotov algorithm must be self-contained. This is currently not the case.

  2. The Authors intermingle the specification of the algorithm with an explanation as to why it is formulated the way it is.

Report

The Authors present a Python implementation of the Krotov quantum optimal control method. The Krotov method is well-known in the optimal control community, and the addition of a high-quality open-source implementation is very welcome and could be of use to a large audience of both optimal control theoreticians and experimentalists. As such, the work is important and should be published.

There are, however, are some issue with the current form of the manuscript which, in my opinion, MUST be amended prior to publication. Primarily, the explanation of the Krotov algorithm in Section 2 requires revising, to make it self-contained as to clearly separate a detailed description of the algorithm from the explanation as to its construction.

13 additional issues which must be addressed (numbered 3-15), and 12 optional comments (numbered 16-27) are listed below.

Requested changes

The explanation of the Krotov algorithm in Section 2 requires revising, on two fronts:

  1. The description of the base Krotov algorithm MUST BE SELF-CONTAINED. For example, the introduction of the Schrödinger equation, a basic feature of the method, is left to references [5,42,43], which puts an unreasonable burden on the reader. While there are many Krotov use cases and variants which can be cited away, one Krotov variant must be presented in full within the manuscript.

  2. The Authors intermingle the specification of the algorithm with an explanation as to why it is formulated the way it is. The result left this reader confused. Please SPECIFY THE ALGORITHM IN A CLEAR, EXPLICIT, CONCISE AND COMPLETE FORM. One possible example is section "Algorithmic Steps" of the cited [20]. The explanation of the algorithm, a self-contained version of Section 2, can then refer to the steps of the algorithm, making it easier for the reader to follow.

Additional issues which, in this reviewer's opinion, MUST be addressed are as follows:

  1. Section 1: The statement "Gradient-based methods typically converge faster, unless the number of optimization parameters can be kept small, and can be separated into methods that update the external control field concurrently or sequentially in time [20]" does not appear to be supported by the data in the cited reference.

  2. Section 2.2: The sentence "It achieves this by adding a vanishing quantity ..." is unclear or incomplete. What is this quantity?

  3. Section 2.2: It is unclear how Eq. (5) follows from Eq. (4).
  4. Section 2.3, point "1. Construct the states ..." The statement "These typically depend on the states..." is unclear. How do they depend on the states? When don't they? It feels as if this explanation is not starting at the beginning.
  5. Section 2.3, point "3. Starting from the known ..." The paragraph is unclear. Please revise. Include Eq. (5) as you wish it to be used. Explain 'This approximation of t ≈ t + dt=2 is what constitutes the "time discretization" mathematically'.

  6. Section 3: "pip install krotov" from Appendix A and the URLs from appendix C belong in section 3.1 or "section 3.0", otherwise one reads about a package without knowing where to get it. The license (BSD, as I discovered on the package's site) should also be noted.

  7. Section 4 and especially 4.3: The comparison made here is limited to the two other methods available in QuTip, GRAPE and CRAB. It is therefore far from a comprehensive comparison of quantum optimal control methods. Therefore, at the very least, the limited scope should be stated clearly - both in the text and in Figure 2.

  8. Section 4.1, "* Krotov’s method mathematically guarantees monotonic convergence in the continuous limit": From the manuscript I understand this to be true only for time-continuous formulations, and explicitly untrue for the time-discrete implementation which is presented here. Therefore, either the manuscript has confused me, or this statement is misleading. Either way something needs to be fixed.

  9. Section 4.1, "Krotov’s method does not require a line search to determine the ideal magnitude of the pulse update in the direction of the gradient.": Aren't we supposed to set \lambda manually? Again, I'm confused.
  10. Section 4.2: "A possible drawback of gradient-free optimization is that it is also prone to get stuck in local optimization minima." - This statement is equally true of many gradient-driven methods, and patently untrue of some gradient-free methods, such as simulated annealing.
  11. Section 4.2: Singling out Subplex in NLopt [60] is unjustified. NLopt, nevergrad [https://github.com/facebookresearch/nevergrad] and other packages contain dozens of relevant algorithms.
  12. Section 4, last paragraph: The text does not clarify why "gradient free" = CRAB, and why the random choice is beneficial.
  13. Section 4, discussion of "time continuous controls": The krotov package explicitly deals with discretized controls. Moreover, it is not clear why the resultant control fields would approximate a continuous control field (i.e. bandwidth constrained), as no smoothing penalty term was demonstrated. And should such a term be introduced, in fairness one must acknowledge a similar term has been used in GRAPE. So it is unclear why the Krotov method should have an advantage in discretized time-continuous controls.

Additional issues, which the Authors may choose to address, are:

  1. Section 1: The paragraph starting with "Large Hilbert space dimensions" - have the authors tried cython?

  2. Section 2, around Eq. (9): Is S(t) multiplied with the control fields? If so, are they bound? If they are unbounded, it is unclear why S(t) would enforce anything; and if they are bounded, the bounding mechanism is unclear.

  3. Section 2.3: The statement "monotonic convergence is mathematically guaranteed" is claimed but not argued. At minimum, repeat citations appearing at the top of Section 2.2.

  4. Section 3.1: How is the use of Krotov with QuTip similar / different than using GRAP or CRAB for the same task? A side-by-side or similar would be useful.

  5. Section 3.1: "* objectives: a list of objectives ..." Why is H per objective? Isn't is identical for all?
  6. Section 3.1: "* propagator: A routine that calculates ..." Why not use the QuTip propagator (I'm guessing there is a good reason - it would be helpful if you specify it).
  7. Section 3.2: "H[1][1]: dict(lambda_a =5, shape =S)": Unclear what is H[1][1] (eps0??) and why is it a key in a dictionary.
  8. Section 3.2: "The value of λa may be adjusted dynamically with respect to the rate of convergence" - How? Which function call?

  9. Appendix A: The discussion of (Ana)conda is extraneous. Python environment management is a mess better left to programming forums.

  10. Appendix B: Moving the complete example to the main text, would greatly help the reader in understanding the calling structure in its entirety. Then special cases can be specified.
  11. Appendix B: The function "hamiltonian.functions def_control" and "S" are almost identical. Isn't there a way to guarantee the required consistency by having "hamiltonian" use "S"?
  12. Appendix B: Several statements in the code are not explained, e.g. "chi_constructor = krotov . functionals . chis_re", "J_T= krotov . functionals . J_T_re".

---

## Round 3 · Referee Report · Nathan Shammah (Referee 2) · 2019-5-30

Strengths

  1. The manuscript and library present a useful tool for quantum optimal control, which is of interest to theoretical and experimental researchers in quantum physics, and it seems particularly timely in the current development of the field of quantum computing and more broadly of quantum information processing, in which noisy intermediate-scale quantum (NISQ) technology devices need to counteract the effect of noise.

  2. The manuscript and software package will make the Krotov method more accessible to a wider audience, increasing the possibility of benchmarking the performance and robustness various gradient-based and gradient-free methods and possibly their interplay.

  3. The library includes cutting-edge features and care toward code optimization. The open-source software package is developed according to the best practices of open-source code development (presentation, online documentation, comments in functions, code formatting, library organization, unit-testing, example notebooks). Useful informative examples are provided online and in the manuscript, as well as code snippets in the manuscript.

  4. Generally, the manuscript is very well written. The authors provide a nice introduction and bibliography for the field of quantum optimal control theory. Discussion of some applications is provided.

Weaknesses

  1. The Introduction section could flesh out more how the Krotov method and its implementation in the presented library can help a researcher in quantum optimal control. General differences with existing methods could be highlighted more, even if some specific comparisons are addressed in detail in the following Section 4 of the manuscript.

  2. The inclusion of graphics could be attractive for the reader. For example, benchmarking and performance plots could be added. Also, plots relative to an example implementation could be included. Some are already present in the online examples, see https://krotov.readthedocs.io/en/latest/08_examples.html.

  3. The discussion of the Krotov method and its implementation in the library are very technical, and there is a somewhat abrupt jump from the general discussion of the Introduction, or the first part of Section 2.1 and, e.g., the detailed description of the terms of Eq. (1). A somewhat more gentle introduction could be appealing to the reader. Also, Section 3 does not look extremely easy to navigate. I do not have a specific recommendation on how to act in this regard, but possibly the introduction of a simple example as that of Appendix B could first gain the attention of the reader in the first part of the manuscript, leaving the full code for the example in the Appendix. In general, a pedagogical approach could greatly benefit the reader and increase the potential user base, whereas currently the discussion becomes quite technical and detail-oriented in Sections 2, 3, and 4. Details are very useful but a bird's-eye view seems somehow missing. This could be quite easily added in the initial parts of these sections to guide the reader with a stronger narrative.

Report

The article provides the implementation of a quantum optimal control method for quantum information processing and quantum technology. The manuscript provides details on the Python library and on the algorithms it implements, providing context on the Krotov method in the quantum optimal control field.

The krotov package enriches a scene in quantum tech research that is increasingly relying on open-source packages to implement investigations and integrate functionalities. The current library seems a timely addition to the toolbox of the quantum tech researcher, at a time when dissipation and noise hamper the ideal implementation of quantum protocols for quantum computing, quantum information processing and quantum control in general.

I am confident that the krotov library will become an important reference point in quantum optimal control theory resources, increasing the popularity of this method and possibly enhancing comparative studies benchmarking various algorithms with open-source code. I hope that the suggested modifications, once implemented in the revised manuscript, will make this tool of even greater help not only for the experts in the field, but for more researchers eager to learn about quantum optimal control with a hands-on approach.

Requested changes

  1. Please consider addressing point (1.) of the weaknesses, i.e. more general comments about the Krotov method and the krotov library in the introductory part of the manuscript.

1.1. For example, in the abstract a line could be added about a defining feature of the Krotov method with respect to other optimal control methods. What about mentioning that Krotov is a gradient-based method?

1.2. Also consider mentioning that Krotov, according to Sec. 2.3 of the paper, assumes time-continuous control fields for assured convergence, but can be applied also for discrete controls.

1.3. With regard to integration with quantum computing experiments, the authors could consider mentioning the advantages of implementing Krotov in currently available quantum computers. If possible, for example, the authors could discuss how would this fare with the OpenPulse scheme announced by IBM, if there are enough details to address it (https://arxiv.org/pdf/1809.03452.pdf). The authors could add if they envision any interaction with existing or future quantum computing platforms, such as IBM, Google, EU's OpenSuperQ.

1.4. With regard to anticipating to the reader a bit more of the contents of the following sections, in the last paragraphs of the Introduction, which includes the structure of the article, more details could be given. Also, information about the Appendices could be included.

  1. Please consider addressing point (2.) of the weaknesses, i.e. adding plots to the manuscript. Also, as an advice, the authors could consider including a table that summarizes already some of the properties of the Krotov method versus other methods.

  2. Consider making Section 2.3 into a pseudo-code which will help the reader to understand the method better and also implement in C++/Julia or any faster language. In section 2.3 (Time discretization) it seems that the core of the implementation is described. This section could be conveniently converted into a pseudo-code of the implementation with some indication of the variable types and explicitly writing down how they should be calculated (not just referencing the equations) e.g.,

"- Construct XT (\xik^{i} (T)), a state vector or density matrix according to Eq (8) as, . math:: \xi_k^{i} (T) = - \frac{\delta J}{\delta \bra {\phi_k }} - Perform a backward propagation over the full-time step using Eq (.) as .. math:: \dot x = - \frac{i}{\hbar} - Store each step in memory (as a list/array of vectors or density matrices)." and so on. This will facilitate an easy understanding of the code behind the implementation, which could help anyone to rewrite the whole tool in a compiled language or otherwise for faster implementation as the authors suggest in the introduction.

  1. Please consider theses additional references:

4.1. On the Krotov method, these references from Ref. [20] seem to be missing and relevant in the discussion: [36] J. P. Palao and R. Kosloff, Phys. Rev. Lett. 89, 188301 (2002). [47] V. F. Krotov and I. N. Feldman, Eng. Cybern. 21, 123 (1983), Russian original: Izv. Akad. Nauk. SSSR Tekh. Kibern. 52 (1983), 162–167. [48] A. I. Konnov and V. F. Krotov, Autom. Remote Control 60, 1427 (1999), Russian original: Avtom. Telemekh. 1999, 77–88.

4.2. Please add information about a recently introduced gradient-based optimal control technique, the GOAT method, "Tunable, Flexible, and Efficient Optimization of Control Pulses for Practical Qubits" Shai Machnes, Elie Assémat, David Tannor, and Frank K. Wilhelm Phys. Rev. Lett. 120, 150401 – Published 9 April 2018 https://doi.org/10.1103/PhysRevLett.120.150401 A discussion of differences and similarities with Krotov would help the interested reader.

4.3. Please consider citing both QuTiP papers when citing that library: J.R. Johansson, P.D. Nation, F. Nori QuTiP: An open-source Python framework for the dynamics of open quantum systems Comp. Phys. Comm. 183, 1760-1772 (2012). J.R. Johansson, P.D. Nation, F. Nori https://www.sciencedirect.com/science/article/pii/S0010465512000835?via%3Dihub

QuTiP 2: A Python framework for the dynamics of open quantum systems Comp. Phys. Comm. 184, 1234 (2013). https://www.sciencedirect.com/science/article/pii/S0010465512003955?via%3Dihub

4.4. On feedback-controlled adiabatic quantum computation: https://journals.aps.org/pra/abstract/10.1103/PhysRevA.86.052306 On efficient quantum control for universal quantum computing: https://journals.aps.org/pra/abstract/10.1103/PhysRevA.81.040303 On quantum feedback control: https://journals.aps.org/pra/abstract/10.1103/PhysRevA.79.052102

  1. About the krotov library: The installation works fine both from conda + pip and by copying the GitHub folder.

5.1. The code snippets of Section 3, at best, should be self-contained.

5.1.1. If one copy pastes them, the code does not run, as many variables are not defined, such as "H", "L", "omega_vals", "delta_vals". Maybe adding these few lines is worth the increased clarity it will provide.

5.1.2. The "itertools" library used on page 11 is not imported or cited in the manuscript.

5.1.3. The example of Appendix B could be hosted online (or in a Zenodo datased; if it is already, it could be said explicitly, when the authors mention the other packages). Note that including the line numbering prevents the reader from easily copy-pasting the code in a script, as the line numbers get copied too.

5.2. In "Krotov.optimize", line 414, the authors could consider to allow the user to separate the exact optimization routine and the other optimization routines. The authors could think of adding other methods beyond negative gradient, see https://arxiv.org/abs/1609.04747. This is just a suggestion, but the authors could address in the manuscript this extension, maybe in the future perspectives of Sec. 5.

5.3. Library testing. 5.3.1. Possible bug report: Running the Example in a Jupyter Notebook, the line

obj.reset_symbol_counters() gave an error AttributeError: 'Objective' object has no attribute 'reset_symbol_counters' Whereas obj.summarize() Provides the expected '|(2⊗2)⟩ - {[Herm[2⊗2,2⊗2], [Herm[2⊗2,2⊗2], u3(t)], [Herm[2⊗2,2⊗2], u4(t)]]} - |(2⊗2)⟩' This is with "module 'krotov' from '/Users/username/miniconda3/lib/python3.6/site-packages/krotov/init.py'"

5.3.2. Possible bug report:

obj.summarize(use_unicode=False) Gave the error summarize() got an unexpected keyword argument 'use_unicode' Although the documentation says this is an option https://krotov.readthedocs.io/en/latest/_modules/krotov/objectives.html#Objective.summarize

From the API docs: https://krotov.readthedocs.io/en/latest/_modules/krotov/objectives.html#Objective.summarize, the krotov.objectives.summarize method has the following signature, "def summarize(self, use_unicode=True, reset_symbol_counters=False)", but in the latest version present in the master branch of the online repository (https://github.com/qucontrol/krotov/blob/master/src/krotov/objectives.py) as well as the pip version 0.30, the summarize method has a different signature,"def summarize(self, ctrl_counter=None)". The respective docstrings are consistent with the change but this is not updated in readthedocs and should be updated perhaps. Adding the Krotov package to Zenodo would provide version release linked to the paper for future reference.

Please double-check the code snippets and examples contained in the manuscript and in the online library documentation using the pip distribution of the software package.

5.4. All of the the Notebooks contained in the GitHub repository online ran fine. The first cell of each requires the user to have the watermark software, which does not get installed with the krotov package from pip, and gives an error. This command could be commented, or the requirement included (beyond for contributing guidelines), or removed from the notebooks or, as a suggestion, a command like qutip.about(), also for the Krotov library, could be added. Also, just as an enquiry, I wonder what "oct" as in "oct_results", used in all notebooks, stands for.

5.5. Although this is not part of the manuscript, all notebooks found on Github ran fine, besides the following minor issues: - notebook 3 is very computationally demanding, and was not run after cell 20;
- notebook 8 required parallel implementation as was partially run; - notebook 9, which gave an error at cell 6 ("Invalid initial_state: must be Qobj, not ndarray"); maybe this modification is part of Krotov development updates.

  1. Detailed comments on the manuscript: 6.1. Introduction 6.1.1. With regard to the sentence: "While GRAPE is found in various software packages, there has not been an open source implementation of Krotov’s method to date." The authors could mention, if they know it, why the Krotov method was not widely implemented so far and what made it appealing to the authors and is appealing to the community in these days.

6.1.2. "Moreover, the QuTiP library [24] exists": Please consider citing both QuTiP papers.

6.1.3. Reference [20], https://arxiv.org/abs/1011.4874, mentions Krotov. From that reference: " We benchmark Krotov-type algorithms and grape algorithms over a selection of scenarios, giving the user of control techniques guidelines as to which algorithm is appropriate for which problem." It seems that this is overlooked in the manuscript.

6.1.4. Regarding the sentence: "Compared to the Fortran and C/C++ languages traditionally used for scientific computing, and more recently Julia [35], pure Python code usually performs slower by two to three orders of magnitude [36,37]. Thus, for hard optimization problems that require several thousand iterations to converge, the Python implementation provided by the krotov package may not be sufficiently fast. In this case, it may be desirable to implement the entire optimization and time propagation in a single, more efficient (compiled) language." What about Cython, would it be of help?

6.1.5. The inclusion of optimization open-systems dynamics, as also detailed in one example, seems very interesting with regard to the realistic model of dynamics in NISQ devices. This feature, which is also present in QuTiP, see https://nbviewer.jupyter.org/github/qutip/qutip-notebooks/blob/master/examples/control-pulseoptim-Lindbladian.ipynb, could be highlighted more in the Introduction.

6.2. Section 1

6.2.1. "The control fields might be the amplitudes of a laser pulse, for the control of a molecular system, RF fields for nuclear magnetic resonance, or microwave fields for superconducting circuits." Control fields for ion-based quantum computing, a promising emerging candidate, could be added.

6.2.2. The DYNAMO package seems to be including Krotov: See https://github.com/shaimach/Dynamo/search?q=krotov&unscoped_q=krotov The presented krotov library is on high level of quality in terms of best practices, documentation, unit testing. Anyhow, consider citing this implementation, which by the way is open source, although it does not rely upon an open source language as Python (is instead based on MatLAB).

6.2.3. Please consider writing explicitly that Eq. (6) is Schrödinger equation and introducing the physics problem in a more gradual way, with more general details.

6.3. Section 2 6.3.1. After Eq. (1), there is a discussion on the functional that is minimized in Krotov. The authors give an example of what the \ket{\phi_k} could look like for two qubit quantum gates. Maybe it will be better to indicate how the guess control function also looks like (is it just a one dimensional function, a pulse?). This is just to keep things pedagogical for the first time reader. Also, the index l seems to be not formally defined. Are there l controls for k states? Or is it independent of the number of states and even for a single state-to-state transfer, there could be more than one (l > 1) controls? As a first-time reader it would be nice to have a self-sufficient picture of the functional to be minimized.

6.4. Section 3 6.4.1. As an advice, the authors can also mention that krotov makes "optimal use" of classes in Python. 6.4.2. Consider spelling once in the paper QuTiP as the Quantum Toolbox in Python. 6.4.3. After equation 17, there is a snippet of code which defines a new sigma class. The S should be in caps as class names in python are recommended to start with caps. 6.4.4. Comment: Very nice flowchart in Figure 2 to select the optimization method based on problem instance. 6.4.5. With regards to "Large Hilbert space dimensions [10,11,33,34] and open quantum systems [29] in particular require considerable numerical effort to optimize.", can we get an idea of the dimensions of the Hilbert space where Krotov starts to face issues - 32? 64? 128? (3 qubits, 5 qubits?) 6.4.6. As a suggestion of In section 3, how to use, installation instruction could be mentioned perhaps in a one-liner. This is discussed in Appendix A and could be cross-referenced here.

6.5. Consider adding a Conclusions section, either adding them to the current Section 5 or dividing the future perspectives from the conclusions of the manuscript, relative to the library, method and applications.

6.6. In Appendix B, which includes a simple example of application of Krotov, the physics should be much better introduced and the results of the notebook should be better presented and more commented. In general, including a general simple example and a more refined high-level example would appeal to the interested coder.

6.7. Referencing: As a suggestion, the authors could consider adding the Krotov package to Zenodo and cite that version release as a crystallized version for future reference; also, Zenodo provides a citeable DOI.

---

## Round 4 · Referee Report · Anonymous (Referee 1) · 2019-10-17

Strengths

  1. The Krotov method is well-known in the optimal control community.

  2. The addition of a high-quality open-source implementation is very welcome and could be of use to a large audience of both optimal control theoreticians and experimentalists.

  3. The manuscript has been significantly revised and improved.

Weaknesses

The manuscript is still short of a paragon of clarity.

Report

The Authors have made significant changes to the manuscript, and as a result it is significantly improved.

It now clears the bar for publication.

No further changes are required prior to publication.

Requested changes

I would suggest the Authors consider tweaking the first page of section 2.2 (the discussion prior to eq. 6), as it is still not as clear as it can be. For example eq. 5 and the surrounding discussion
would probably be better if they appear after eq. 8. This is optional, and left to the Authors' discretion.

  • validity: high
  • significance: good
  • originality: ok
  • clarity: good
  • formatting: excellent
  • grammar: excellent

Author:  Michael Goerz  on 2019-12-05  [id 666]

(in reply to Report 2 on 2019-10-17)

We greatly appreciate and agree with the referee's suggestion to improve the clarity of the manuscript. In the final revision (<https://arxiv.org/abs/1902.11284v5>), we have rearranged the discussion, and split Section 2.2 into a new Section 2.2 and 2.3.

---

## Round 4 · Referee Report · Nathan Shammah (Referee 2) · 2019-10-24

Report

The authors have greatly improved the already important manuscript, beginning from the abstract, from content extensions to clearer explanations, also thanks to an improved structure of the paper, and a stronger and more engaging narrative.

I appreciate their assessment of the points raised and I agree with their replies to my comments and questions. I am glad that point 6.4.4 and other suggestions have been implemented, to the benefit of the paper, I believe.

Section 2.3 considerably improves the clarity of the paper. The Conclusions make the article's perception more impactful than in the previous version.

Therefore, I strongly advise the paper for publication in SciPost Physics.

  • Comments on replies and new comments*:
  • I am convinced by all points made by the authors, and in retrospect also with 6.7 on the ambiguity of an additional DOI with Zenodo: pip versioning will serve the scope of code "versioning".

  • A comment on why Blackman shape is a popular choice might be nice.

  • When discussing Fig. 1, the methods (or attributes) on opt_dynamics and else used to plot both the converging and final results of the populations and control pulses should be mentioned explicitly for clarity.

  • I think $N$ is used in different ways in Sec. 2.2 and Sec. 3.3, this could be uniformed.

  • In the tree of Fig. 2 and main text, the fact that $n$ could be explicitly defined as the number of control parameters.

When I read in the first bullet point of Sec. 4.4 that "Some potential benefits of Krotov’s method compared to GRAPE are [69]: • Krotov’s method mathematically guarantees monotonic convergence in the continuous- time limit. There is no line-search required for the step width 1/λa,l." it seemed like a big deal, a bit hidden at page 20. This fact is indeed mentioned in the Introduction, "A popular representative of concurrent update methods is GRadient Ascent Pulse Engineering (GRAPE) [21], whereas Krotov’s method, which comes with the advantage of guaranteed monotonic convergence, requires sequential updates". It is only a suggestion, but I think that even only reshuffling the syntax of this introductory sentence may make this very nice feature stand out a bit more, engaging more the reader. Also non-experts in quantum optimal theory may know of GRAPE, so this may be a point to underscore when introducing the lesser known Krotov method. Just a suggestion.

Response to the Reply to 5.2: In section 4.2 of the new manuscript the authors actually mention something similar to what I had in mind with that comment:

"In all its variants [5,22,46–48], Krotov’s method is a first-order gradient with respect to the control fields (even in the second-order construction which is second order only with respect to the states). As the optimization approaches the optimum, this gradient can become very small, resulting in slow convergence. It is possible to extend Krotov’s method to take into account information from the quasi-Hessian [69]. However, this “K-BFGS” variant of Krotov’s method is a substantial extension to the procedure as described in Appendix B, and is currently not supported by the krotov package."

I simply meant that it could be possible to have such kind of extensions (K-BFGS) where the user could choose addition techniques to get some advantage while running Krotov.

As they mention, it is beyond the scope of this paper and was merely a comment.

The code snippet in Section 2 run fine on my Mac. Just a suggestion, or idea: in the table that is printed when running krotov.optimize_pulses, words could define the quantities at the beginning of the columns. Like, over "J_T" have "final t. functional", in the next column "running cost", or similar, and so on. For an experienced krotov user this is trivial, but maybe for a newcomer it would help grasp what's happening.

Minor comments: - The Liouvillian is not explicitly introduced in Eq. (3), and it could be mentioned it is not a Hermitian matrix. - Just as a suggestion, maybe the initial and target states expressions, present in Eq. (6), could be introduced in the relevant main text discussion occurring after Eq. (3), for redundancy between the general concept of optimal control and Krotov's.
- Now, after the equation reshuffling of Sec. 2.1, the shape function $S(t)$ is not defined when it is first introduced, immediately after Eq. (8). - the small equal sign under the braces is a bit confusing and maybe not necessary
- The frequency $\omega$ and control pulse $\epsilon(t)$ could be introduced in Section 2.3 also before the code block, even if the control pulse concept has been introduced in the previous sections. - After Eq. (12) on page 10, it would be clearer to cite Results as qutip.Results, otherwise it seems the Results are a krotov class. I am a bit confused here, because looking at the code, there is a result.py file and a Result class defined therein, but running help(krotov.Result) prompts an error message; indeed help(opt_dynamics) tells it's a qutip.Result object from the mesolve dynamics.

  • In references, Ref. 14 is missing volume and page, Ref. 29 "rydberg" needs capital R.

Just an idea (not something to address in resubmission): the print message during the optimization, instead of having start and end date and time, could just report the total running time.

  • validity: -
  • significance: -
  • originality: -
  • clarity: -
  • formatting: -
  • grammar: -

Author:  Michael Goerz  on 2019-12-05  [id 665]

(in reply to Report 1 by Nathan Shammah on 2019-10-24)
Category:
reply to objection

We thank the reviewer for another detailed and constructive review, and for their recommendation to publish.

We have revised the manuscript on the ArXiV (https://arxiv.org/abs/1902.11284v5) addressing all remaining issues according to the reviewer's suggestion. The full list of changes (including those in response to the second referee report) is the following:

  • Changed URL for package documentation to https://qucontrol.github.io/krotov, which currently forwards to Read-The-Docs, but will allow us to move to a different documentation hoster in future without invalidating the link in the paper.
  • Added sentence to emphasize the benefit of a Blackman shape (zero at beginning and end) more strongly.
  • Added a sentence pointing to the online documentation for details on how Fig 1 is generated.
  • Switched index N in Section 3.3 to M to avoid confusion with N in section 2.2
  • Added definition of "n" as the number of control parameter in Fig 2 and text.
  • Rephrased introductory sentence to highlight monotonic convergence without the need of a line search
  • Corrected capitalization of "Rydberg" in references.
  • Added definition of "Liouvillian" below Eq. (3), and specify that it is non-Hermitian
  • Added the initial and target states expressions in the text after Eq. (3)
  • Added definition of the "inverse step width" and "update shape" function immediately below Eq. (8) [now Eq. (7)]
  • Removed equal signs from underbraces
  • Added Definition of symbols ω and ϵ at beginning of Section 2.3 [now Section 2.4]
  • Restructured Section 2.2 for clarity, separating the discussion of the functional from the discussion of the iterative pulse update in a new Section 2.3
  • Added two missing references for GROUP-like methods, and one reference for QEngine (implementation of GROUP)
  • Reformatted the "input" for Algorithm 1 as an enumeration, for better readability
  • Small rephrasing in the Introduction: split the sentence "A popular representative of concurrent update methods is GRadient Ascent Pulse Engineering (GRAPE), whereas Krotov's method, which comes with the advantage of guaranteed monotonic convergence, requires sequential updates" into two sentences
  • Update installation instructions to use Python 3.7 (the latest release of Python officially supported by krotov/QuTiP), instead of Python 3.6.

As a further comment towards specific points:

  • A comment on why Blackman shape is a popular choice might be nice.

In this case, only the fact that the Blackman shape goes exactly to zero at the beginning and end is important. Without that, in each iteration, the initial and final value of the control might shift away from zero very slightly.

  • When discussing Fig. 1, the methods (or attributes) on opt_dynamics and else used to plot both the converging and final results of the populations and control pulses should be mentioned explicitly for clarity.

We now mention the attribute involved, but the full details (including the somewhat lengthy code to generate the plot) is best presented in the corresponding notebook in the online documentation (which we now point to in the paper).

  • I think N is used in different ways in Sec. 2.2 and Sec. 3.3, this could be uniformed.

Indeed, and we have now switched N to M in Sec 3.3.

The code snippet in Section 2 run fine on my Mac. Just a suggestion, or idea: in the table that is printed when running krotov.optimize_pulses, words could define the quantities at the beginning of the columns.

We might consider this in a future version. For now, the output matches exactly what users of our existing Fortran QDYN package are used to.

Just an idea (not something to address in resubmission): the print message during the optimization, instead of having start and end date and time, could just report the total running time.

We have added the running time in the latest release of the krotov package in addition to the start/end time stamps.

---

## Round 4 · Author Response

We thank both referees for their thorough and constructive comments and have revised and extended the manuscript substantially according to their suggestions.

We respond to the specific requested changes in the first referee's report as follows:

1.-2.: We agree that the manuscript should be self-contained in specifying the algorithm implemented in the package in full detail, to the point that the reader could implement Krotov's method on their own. However, it is not the aim of the paper to re-derive Krotov's method in full. This derivation is both lengthy and highly technical and is best found (apart from Krotov's original papers) in the cited Reich et al, J. Chem. Phys. 136, 104103 (2012).
We have restructured the manuscript to put more emphasis on the *usage* of the package, and to keep more technical aspects such as the discussion of Krotov's update equation and the full description of the algorithm in the Appendix. This now includes a complete pseudocode specification of the optimization method in Appendix B. This augments the less technical description of the algorithm in Appendix A3. Also, we now introduce the Schrödinger and Liouville equations explicitly at the beginning of Section 2.

3.: Reference [20] was meant to relate to the distinction between sequential and concurrent update schemes only. We have split the sentence to clarify this and generally revised the discussion.

4.: We agree that this sentence is confusing outside of the full derivation of Krotov's method (which is out of scope for the current manuscript). We have removed it. As a clarification to the referee: we were referring to Phi in Eq (7) of ReichJCP2012.

5.: We have added Eq. (26) in the manuscript that explains how the update equation follows from the choice of $g_a$.

6.: We have revised this section to specify that the chi-states depend on the forward-propagated states if they are more than linear in the overlaps for forward-propagated states and target states.

7.: We have written out the explicit time-discretized update equation, and revised the discussion.

8.: We now reference Appendix C before discussing the example, and mention the BSD license in Appendix D.

9.: Our scope is that of open-loop control methods, but within that scope, our aim is to give a discussion that is complete with respect to all methods that we are aware of. However, especially in Fig. 2, each method is meant to include a significant number of variants. We have extended the discussion of the different methods significantly. We have also included a discussion of the recent GROUP/GOAT methods.

10.: Monotonic convergence is still expected in the discretized scheme, assuming the controls are discretized from continuous fields with a time step sufficiently small that the discretization error in the propagation is negligible. It may require a sufficiently large value for the step width lambda: values that are too small may lead to numerical instabilities where monotonic convergence is lost, but typically this is easily detected (and, the resulting optimized controls would be "spiky", violating the requirement that the discretized controls must be a good approximation to a continuous fields. We have added this dependency on lambda to the text.

11.: The line search in GRAPE is performed automatically in each iteration to determine how far to change the field in the direction of the gradient. Without a line search, the algorithm will not converge. In contrast, the parameter lambda does not require a line search. In the continuous limit, monotonic convergence is guaranteed for *any* value of lambda. In the discretized scheme, the only limit on lambda is that very small values will lead to numerical instability. Beyond that, the choice of lambda has some effect on the smoothness of the optimized fields: the less the controls change in each iteration (large lambda), the smoother the result, typically. In practice, lambda is usually set at the beginning of the optimization and kept unchanged. Alternatively, as we discuss, one may start with a large value of lambda the first few iterations to counteract the typically large changes in the first few iterations, and then lower the value by an order of magnitude or so after that. However, it is not common to change lambda in each iteration, and we believe that doing so (by a line search) is unlikely to give an improvement that is worth the additional cost of the line search.

12.: We concur with the referee and have removed the paragraph.

13.: We have revised the section, and included a reference to Nevergrad.

14.: We did not mean to imply that CRAB is the only relevant gradient-free method. We have revised the discussion accordingly.

15.: While Krotov's method in its standard form does not contain bandwidth restrictions that would guarantee smoothness, assuming the guess pulses are approximately continuous, it is an inherent feature of the method that the optimized pulses will also be continuous. This is up to numerical instabilities due to a choice of a lambda that is too small (which is easily detected). That being said, in practice GRAPE can typically be applied to approximately time-continuous pulses as well without problems, so this is largely a matter of personal preference. The advantages of Krotov are those listed in section 4.3 (monotonic convergence, faster initial convergence, and no line search).

16.: We are currently exploring the use of Cython for an efficient propagator that we hope will extend the usability of the krotov package to significantly larger Hilbert space dimensions.

17.: The are no inherent bounds on the control field (bounds may be introduced through parametrization, as outlined in Section 5). The function S(t) is not multiplied with the control field, but with the *update* in each iteration. Thus, the only bound it may place on the optimized fields is that the optimization will not change the field amplitude for any value of t where S(t) is zero. In particular, S(t) commonly is zero at t=0 and t=T, which preserves the original values of the guess control at t=0 and t=T (which often are also zero). We have renamed the corresponding parameter from "shape" to "update_shape" in the package, and rewritten the discussion to clarify this.

18.: We have revised the section to explain the origin of the monotonic convergence.

19.: We agree with the referee that a comparison of different optimization algorithms applied to the same physical problems is interesting, but it is beyond the scope of the current paper. Moreover, the current implementation of GRAPE and CRAB is limited and currently does not allow for arbitrary time-continuous guess controls. A comparison would thus require extending QuTiP's capabilities. We note that there are some comparisons of the various method in the cited literature, e.g. Schirmer et al. New J. Phys. 13, 073029 (2011), Machnes et al. Phys. Rev. A 84, 022305 (2011), Jäger et al. Phys. Rev. A 90, 033628 (2014), and Sørensen et al. Phys. Rev. A 98, 022119 (2018).

20.: In most situations, H is indeed identical in all objectives. However, this is not a requirement. The particular example where H is different in each objective is the "ensemble optimization" discussed in Section 3.3. The idea here is to consider the same process under different Hamiltonians, each with a different set of "noise" parameters. By optimizing over all objectives (noise Hamiltonians) simultaneously, the result will ideally be "robust".

21.: We require the "propagator" to simulate the time dynamics over a single time step. The mesolve routine in QuTiP only supports propagation over an entire time grid. The overhead of restarting mesolve in each step would be prohibitive. We have experimented with refactoring the internals of mesolve into a more suitable form, but this is not without challenges.

22.: H[1][1] is the guess_control from the "hamiltonian" function. We have added a comment to explain this.

23.: The value of lambda may be modified in a callable passed as "modify_params_after_iter" to "optimized_pulses". We have clarified this in the text.

24.: We do not agree that mentioning Anaconda as a recommendation is problematic. QuTiP is non-trivial to install, and in our opinion Anaconda currently provides the easiest way to do so on systems that do not have the necessary compilers installed to build extension modules.

25.: We have restructured the manuscript according to the referee's suggestion.

26.: The shape (envelope) of the control field and the "update shape" that scales the update at each point in time, are not related. We have revised the krotov package, and the paper, in an attempt to make this more explicit.

27.: We have revised the discussion to explain all arguments to the optimized_pulses function, and chi_constructor in particular

In response to the second referee's requested changes,

1.: We have significantly expanded the comparison of Krotov's method and other relevant optimization methods.

1.1: We have added a sentence to the abstract defining Krotov's method as a gradient-based method, according to the referee's suggestion.

1.2: We have revised the discussion of Krotov's method substantially in a way that should clarify time discretization.

1.3: We are very interested in exploring the possibility of integrating Krotov's method with current quantum cloud-computing libraries. However, at this time we have insufficient experience with these libraries to enter into a meaningful discussion. We hope to rectify this in future publications.

1.4 We have revised the last paragraph of the introduction to include the information in the appendices.

2.: We have added a plot to the manuscript illustrating the result of the example. As the numerical effort is mainly in the propagation routine (which in general should be user-supplied), not in the implementation of Krotov's method, performance benchmarks would not be very meaningful. We may reconsider benchmarks once a Cython-based single-time-step propagator becomes available.

3.: We have added an appendix containing the full pseudocode for the method, augmenting the high-level description of the algorithm.

4.1: We have added the references to the manuscript.

4.2: We have added a discussion of the GROUP/GOAT method.

4.3: We now cite both QuTiP papers.

4.4: The scope of our discussion is only open-loop control schemes, which we have clarified in the text.

5.1: Some of the code snippets require considerable setup. We feel that including this additional code would detract from the clarity of the code snippets. However, the full scripts from which the snippets originate are available online as part of the package's source code repository. This is mentioned in the text and in Appendix D.

5.1.1: The code snippets are not intended to be run by copy/paste. The complete example script of Section 2.3 is available online. For the shorter snippets from Section 3, complete scripts are also available online, although we would recommend a reader to experiment with the equivalent interactive notebooks in the online documentation instead. The script files for the shorter snippets are intended mostly for automated testing, to verify their correctness.

5.1.2: "itertools" is a standard library package. We have clarified this with a comment.

5.1.3: Following the referee's recommendation, the example script is now available online

5.2: We are not quite clear what the referee has in mind here. In particular, the various extensions to gradient-descent like stochastic gradient descent or the inclusion of a "momentum" do not appear to be relevant to Krotov's method.

5.3: These errors would seem to come from a mismatch of example notebook based on the most recent released version of the krotov package, and subsequent modifications on the master branch. We have modified the online documentation to show the latest released version by default, and to show a warning when a user is viewing documentation for an unreleased development version. To the extent that any of these problems are still present in the most recent version of the krotov library, we would ask the reviewer to open an issue on Github.

5.4: The example notebooks use a variety of packages (like "watermark") that are not dependencies of the krotov package. This is by design. However, the dependencies for the notebooks are included in the package's "development" requirements, which may be installed through "pip install krotov[dev]". We have added this to the manuscript and will address it in the online example notebooks.

"oct" in the name of the variables is from "Optimal Control Theory", and has slipped through as an acronym that we commonly use in internal code. However, it is admittedly jargon, and we have replaced "oct" with "opt", as in "opt_result" for the optimization result.

5.5.: We cannot reproduce these issues in the most current version of the package. To the extent that this is still a problem, we would ask the reviewer to submit an issue to Github.

6.1.1: There is no fundamental reason Krotov's method hasn't been implemented so far, just that the number of groups with sufficient expertise to do so is relatively small, and nobody has gotten around to it until now.

6.1.3: The DYNAMO library implements a method we would call "sequential GRAPE", which some parts of the quantum control community label "Krotov-type". It is quite different from the method we present here. We have added a detailed discussion of this, and specified Krotov's method in far greater detail.

6.1.4: Cython would certainly be an option. In particular, it may be useful to improve the parallelization of the Krotov implementation by releasing the global-interpreter-lock. However, apart from parallelization, we suspect that the most effective place for Cython would be in implementing an efficient propagator. We are currently working on this. The numerical effort of the optimization is dominated by the propagation, and it is not clear whether cythonizing Krotov further than that will lead to a substantial improvement. We will explore this in more detail in the future.

6.1.5: We have added a sentence about open quantum systems and NISQ in the Introduction.

6.2.1: We now mention trapped ion quantum computers in the Introduction.

6.2.2: The DYNAMO package does not implement Krotov's method, but a "Krotov-style" version of sequential gradient ascent (see 6.1.3).

6.2.3: We now introduce the Schrödinger equation in more detail.

6.3.1: We discuss the control fields in the new section 2.1. This should also clarify the role of the index "l". Indeed, the number of controls is not related to the number of states, but to the physically independent control fields. In the case of complex-valued control, e.g. in a rotating frame, there are two controls (real and imaginary part) for each physical control.

6.4.1: We are not sure what constitutes "optimal use of classes", although we appreciate the assessment.

6.4.2.: We have spelled out the name for QuTiP in the revised version of the manuscript.

6.4.3.: We are aware of the usual conventions, but we feel that in numerical code it is sometimes best to relax them. Thus, we deem capitalized single-letter or Greek variable names acceptable if they correspond to uppercase mathematical variables, or lowercase class names for classes that represent lowercase mathematical variables (as is the case here).

6.4.4: We appreciate the referee's comment. We have extended the flowchart to include a wider range of methods.

6.4.5: The size of the Hilbert space that is still numerically feasible is largely determined by the efficiency of the propagation routine. Based on our experience with a Fortran implementation of Krotov's method, Hilbert space dimensions greater than 10000 become challenging. For the krotov package with a propagator in pure Python, this limit will be significantly lower, probably around 100. We are not sure yet how far the limit can be pushed by implementing the propagator in Cython.

6.4.6: We now mention the Appendix with the installation instructions early in the paper.

6.5: We have added a Conclusions section.

6.6: We have moved the example to the main text and expanded the discussion. For more advanced examples, we would refer the reader to the online documentation.

6.7: We intend the manuscript as the "citation object" for use of the package, and feel that assigning a DOI to a specific release of the package might compete with that.

---

## Round 4 · List of Changes

We have thoroughly revised the manuscript in its entirety, see the response to the referee reports.

---

## Editorial Decision

published